# SWING: Unlocking Implicit Graph Representations for Graph Random Features

**Alessandro Manenti** [* 1]   **Avinava Dubey** [2]   **Arijit Sehanobish** [3]   **Cesare Alippi** [1 4]   **Krzysztof Choromanski** [* † 5]

## Abstract

We propose SWING: Space Walks for Implicit Network Graphs, a new class of algorithms for computations involving Graph Random Features (Choromanski, 2023) on graphs given by implicit representations (i-graphs), where edge-weights are defined as bi-variate functions of feature vectors in the corresponding nodes. Those classes of graphs include several prominent examples, such as: $\epsilon$-*neighborhood* graphs, used on regular basis in machine learning. Rather than conducting walks on graphs' nodes, those methods rely on walks in continuous spaces, in which those graphs are embedded. To accurately and efficiently approximate original combinatorial calculations, SWING applies customized Gumbel-softmax sampling mechanism with linearized kernels, obtained via random features coupled with importance sampling techniques. This algorithm is of its own interest. SWING relies on the deep connection between implicitly defined graphs and Fourier analysis, presented in this paper. SWING is accelerator-friendly and does not require input graph materialization. We provide detailed analysis of SWING and complement it with thorough experiments on different classes of i-graphs.

## 1. Introduction & Related Work

Graphs are ubiquitous in machine learning (ML), representing prominent data modality that provides important structural inductive bias in the form of pairwise relationships between objects represented as their nodes/vertices. For that reason, graphs have found several applications in various ML applications benefitting from rich relational signal coming from data. Those range from molecular modeling (drug discovery, property prediction, protein-ligand modeling; see: (Subedi et al., 2024; Sypetkowski et al., 2024; Soares et al., 2025; Kearnes et al., 2016; Yang et al., 2022; Méndez-Lucio et al., 2022; Borgwardt et al., 2005; Fober et al., 2013; Renaud et al., 2020)), through recommender systems (Deng et al., 2022; Wei et al., 2022; Xu et al., 2024) and manifold learning (Lim et al., 2024; Hein & Maier, 2006; Trillos et al., 2023) to 3D modeling for point clouds and robotics (Choromanski et al., 2023; Lin et al., 2021; Zhang et al., 2023; Zheng et al., 2025).

Graph data provides several advantages over more standard modalities, but this comes with the cost. It is well illustrated by kernel methods applied for graphs. Similarity functions (kernels) between nodes in the graphs or the graphs themselves (Kondor & Lafferty, 2002) are much harder to compute than regular kernels (e.g. Gaussian) in Euclidean spaces. In particular, computing kernel-matrices for several classes of well-established graph kernels, e.g. diffusion kernels between graph-nodes or random-walk kernels between graphs, takes time $O(N^3)$ or even $O(N^6)$ (if the similarity measure is lifted from the graph's nodes-space to graphs' space) (Vishwanathan et al., 2010; Neuhaus & Bunke, 2006), where $N$ is the number of vertices. Thus in practice applying those techniques is constrained to relatively small graphs of a few hundred nodes.

To address this challenge, a class of *Graph Random Features* methods (or GRFs) was recently proposed (Choromanski, 2023; Reid et al., 2024). GRFs provide an unbiased factorization or the graph kernel matrices into sparse matrices by considering random walks on the input graphs. Those sparse matrices can be in turn mapped into low-rank (dense) matrices, still providing unbiased approximation. Consequently, GRFs can significantly improve computational efficiency of the graph kernel methods and scale up their applicability to graphs of tens of thousands of nodes and larger, by leveraging efficient calculations with sparse and dense low-rank matrices. GRFs were originally designed for kernels between graphs' nodes, but were also extended to kernels between graphs (Choromanski et al., 2025b). In a related graph-level setting, other randomized feature constructions were also explored (Zambon et al., 2020).

GRFs were already successfully used in a large volume of applications, such as: clustering algorithms with graph kernels (Choromanski, 2023), differential equations on graphs,

*Equal contribution   [1]Università della Svizzera italiana, IDSIA [2]Google Research [3]Independent Researcher [4]Politecnico di Milano [5]Google DeepMind and Columbia University. [†]Senior lead   Correspondence to: Alessandro Manenti <alessandro.manenti@usi.ch>, Krzysztof Choromanski <kchoro@google.com>.

*Proceedings of the $43^{rd}$ International Conference on Machine Learning*, Seoul, South Korea. PMLR 306, 2026. Copyright 2026 by the author(s).

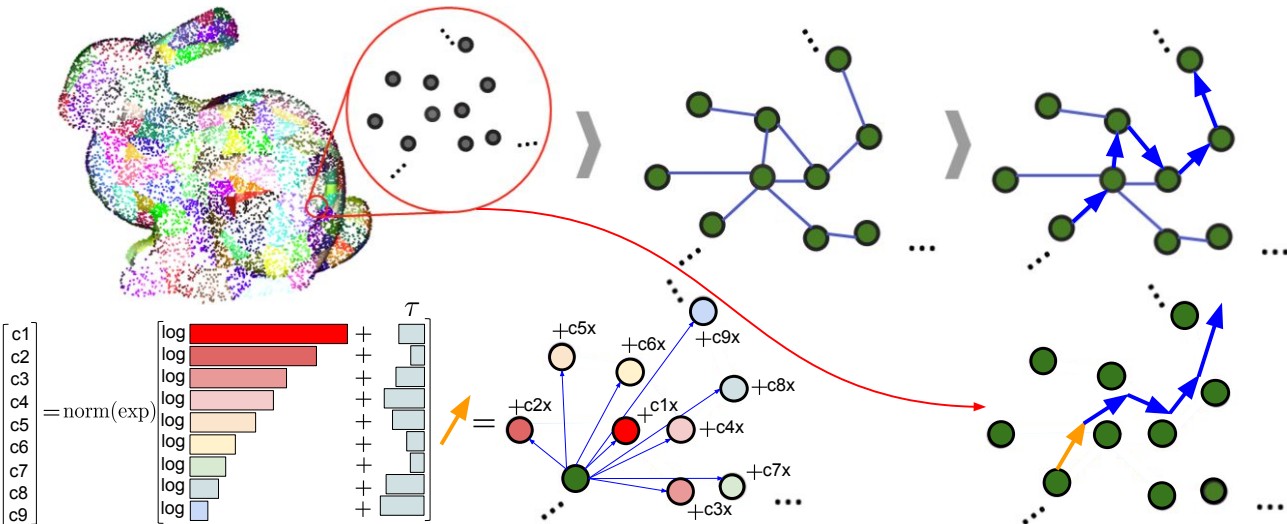

*Figure 1.* Pictorial visualization of the SWING mechanism on the example of the i-graph obtained from the point cloud. The scheme above the red line shows an approach based on regular Graph Random Features: an explicit graph representation with nodes and weighted edges is constructed and used to conduct walks (here depicted via blue arrows) on the nodes of the constructed graph. SWING bypasses this step by conducting those random walks in the continuous $d$-dimensional space, where i-graph's nodes are embedded (see: scheme to which red arrow is pointing). Each transition in such a walk (see: orange arrow) is obtained as a convex sum of the vectors joining current location of the walker with the locations of the nodes of the graph. The coefficients of that convex sum are proportional to the exponentiated sums of the weight-logarithms and Gumbel variables. That turns out to provide differentiable relaxation of the discrete random walk.

node attribute prediction (such as vertex-normals on meshes) (Reid et al., 2024), 3D-Transformers (Reid et al., 2025) and very recently: Gassian processes on graphs (Zhang et al., 2025) and graph classification (Choromanski et al., 2025a).

Despite their universality, GRFs come with one major limitation. They require explicit graph construction for random walks. This is problematic since combinatorial operations such as walks on graphs are not well-supported by modern accelerators. Furthermore, in several ML applications input graphs of interest are not even provided explicitly, but rather *implicitly* (from no on, we will refer to them as *i-graphs*). The weights of edges $w_{i,j}$ of those graphs are defined as follows:

$$w_{i,j} = f(v(i) - v(j)) \quad (1)$$

for some function $f : \mathbb{R}^d \to \mathbb{R}$ and mapping $v : V \to \mathbb{R}^d$ from the set of vertices to their corresponding feature vectors. The classic example constitue $\epsilon$-*neighborhood* graphs, for which $f$ is given as: $f(\mathbf{x}) = \mathbb{I}(\|\mathbf{x}\| \leq \epsilon)$ and often $d = 3$ (examples include point cloud modeling, where vectors $v(i)$ correspond to 3D-coordinates of points). Here $\mathbb{I}$ stands for the indicator function.

Thus, even if explicit representations of such graphs (that GRFs operate on) are of quadratic complexity in the number of vertices $N$, those graphs by definition provide computationally-efficient implicit representations. They are not leveraged by GRFs though.

In this paper, we propose SWING: **S**pace **W**alks for **I**mplicit **N**etwork **G**raphs, a new class of algorithms for compu-

tations involving Graph Random Features (Choromanski, 2023) on i-graphs. To be more, specific, SWING supports linear-time approximate multiplications with kernel matrices induced by GRFs, and **without graph materialization**. Rather than conducting walks on graphs' nodes, those methods rely on walks in continuous spaces, in which those graphs are embedded. To accurately and efficiently approximate original combinatorial calculations, SWING applies customized Gumbel-softmax sampling mechanism with linearized softmax-kernel (see: Fig. 1), obtained via random features coupled with importance sampling techniques. This algorithm is of its own interest. SWING relies on the deep connection between implicitly defined graphs and Fourier analysis, presented in this paper. SWING is accelerator-friendly and does not require input graph materialization. Its time complexity is $O(N)$ for i-graphs, even if they have $\Theta(N^2)$ edges. We provide detailed analysis of SWING and complement it with thorough experiments on different classes of i-graphs.

## 2. The Algorithm

### 2.1. Preliminaries: Graph Random Features

SWING provides a relaxation of the original combinatorial computational mechanism of Graph Random Features (GRFs). Thus we introduce GRFs first.

We will consider weighted undirected N-vertex graphs $G(V, E, \mathbf{W} = [w_{i,j}]_{i,j \in V})$, where (1) V is a set of vertices, (2) $E \subseteq V \times V$ is a set of undirected edges ($(i, j) \in E$

**Algorithm 1 Regular GRFs (for weighted graphs):** Construct vectors $\xi_\rho(i) \in \mathbb{R}^N$ to approximate $\mathbf{K}_\alpha(\mathbf{W})$

**Input:** weighted adjacency matrix $\mathbf{W} \in \mathbb{R}^{N \times N}$, vector of unweighted node degrees (number of out-neighbours) $\deg \in \mathbb{R}^N$, modulation function $\rho : (\mathbb{N} \cup \{0\}) \to \mathbb{R}$, termination probability $p_{\text{halt}} \in (0, 1)$, node $i \in \mathcal{N}$, number of random walks to sample $m \in \mathbb{N}$.

**Output:** signature vector $\xi_\rho(i) \in \mathbb{R}^N$.

1: initialize: $\xi_\rho(i) \leftarrow \mathbf{0}$
2: **for** $w = 1, ..., m$ **do**
3:    initialize: load $\leftarrow 1$, current_node $\leftarrow i$, terminated $\leftarrow$ False, walk_length $\leftarrow 0$
4:    **while** terminated = False **do**
5:       $\xi_\rho(i)[$ current_node $] \leftarrow \xi_\rho(i)[$current_node$] +$ load $\times \rho($ walk_length $)$
6:       walk_length $\leftarrow$ walk_length $+ 1$
7:       new_node $\leftarrow$ Cat $[\mathcal{N}($ current_node $)]$ ▷ assign to one of neighbours, with probabilities $\sim$ to edge weights
8:       load $\leftarrow$ load $\times \frac{\deg[\text{current\_node}]}{1-p_{\text{halt}}}$ ▷ update load; $\deg(v) \overset{\text{def}}{=} \sum_{z \in \mathcal{N}(v)} \mathbf{W}(v, z)$
9:       current_node $\leftarrow$ new_node
10:      terminated $\leftarrow (t \sim \text{Unif}(0, 1) < p_{\text{halt}})$ ▷ draw RV $t$ to decide on termination
11:    **end while**
12: **end for**
13: normalize: $\xi_\rho(i) \leftarrow \xi_\rho(i)/m$

---

indicates that there is an edge between $i$ and $j$ in G), and (3) $\mathbf{W} \in \mathbb{R}_{\geq 0}^{N \times N}$ is a weighted adjacency matrix (no edge results in the zero weight).

We consider the following kernel matrix $\mathbf{K}_\alpha(\mathbf{W}) \in \mathbb{R}^{N \times N}$, where $\boldsymbol{\alpha} = (\alpha_k)_{k=0}^\infty$ and $\alpha_k \in \mathbb{R}$:

$$\mathbf{K}_\alpha(\mathbf{W}) = \sum_{k=0}^\infty \alpha_k \mathbf{W}^k. \tag{2}$$

For bounded $(\alpha_k)_{k=0}^\infty$ and $\|\mathbf{W}\|_\infty$ small enough, the above sum converges. The matrix $\mathbf{K}_\alpha(\mathbf{W})$ defines a kernel on the nodes of the graph. Interestingly, Eq. 2 covers as special cases several classes of graph kernels, in particular graph diffusion/heat kernels.

GRFs enable one to rewrite $\mathbf{K}_\alpha(\mathbf{W})$ (in expectation) as $\mathbf{K}_\alpha(\mathbf{W}) \overset{\mathbb{E}}{=} \mathbf{K}_1 \mathbf{K}_2^\top$, for independently sampled $\mathbf{K}_1, \mathbf{K}_2 \in \mathbb{R}^{N \times d}$ and some $d \leq N$. This factorization provides efficient (sub-quadratic) and unbiased approximation of the matrix-vector products $\mathbf{K}_\alpha(\mathbf{W})\mathbf{x}$ as $\mathbf{K}_1(\mathbf{K}_2^\top \mathbf{x})$, if $\mathbf{K}_1, \mathbf{K}_2$ are sparse or $d = o(N)$. This is often the case in practice. However, if this does not hold, explicitly materializing $\mathbf{K}_1 \mathbf{K}_2^\top$ enables one to approximate $\mathbf{K}_\alpha(\mathbf{W})$ in quadratic (versus cubic) time. Below, we describe the base GRF method for constructing sparse $\mathbf{K}_1, \mathbf{K}_2$ for $d = N$. Exten-

sions giving $d = o(N)$, using the Johnson-Lindenstrauss Transform (Dasgupta & Gupta, 2003), can be found in (Choromanski, 2023). Each $\mathbf{K}_j$ for $j \in \{1, 2\}$ is obtained by row-wise stacking of the vectors $\xi_\rho(i) \in \mathbb{R}^N$ for $i \in V$, where $\rho$ is the *modulation function* $\rho : \mathbb{R} \to \mathbb{R}$, specific to the graph kernel being approximated. The procedure to construct random vectors $\xi_\rho(i)$ is given in Algorithm 1. Intuitively, one samples an ensemble of RWs from each node $i \in V$. Every time a RW visits a node, the scalar value in that node is updated by the renormalized *load*, a scalar stored by each walker, updated during each vertex-to-vertex transition. The renormalization is encoded by the modulation function.

After all the walks terminate, the vector $\xi_\rho(i)$ is obtained by the concatenation of all the scalars from the discrete scalar field, followed by a simple renormalization. For unbiased estimation, $\rho : \mathbb{N} \to \mathbb{C}$ needs to satisfy $\sum_{p=0}^k \rho(k-p)\rho(p) = \alpha_k$, for $k = 0, 1, ...$ (see Theorem 2.1 in (Reid et al., 2024)). We conclude that modulation function $\rho$ is obtained by deconvolving sequence $\boldsymbol{\alpha} = (\alpha_k)_{k=0}^\infty$ defining graph kernel.

**Note:** We would like to emphasize that matrix $\mathbf{W}$ can be also a derivative of the regular weighted adjacency matrix, e.g. obtained by conducting degree-based renormalization, as in (Reid et al., 2024).

### 2.2. Space Walks on Implicit Network Graphs

We are ready to present SWING algorithm. We assume that the vertices of the $N$-vertex graph are embedded in the $d$-dimensional space, with the corresponding feature vectors: $v(1) = \mathbf{p}_1, ..., v(N) = \mathbf{p}_N \in \mathbb{R}^d$.

#### 2.2.1. REPLACING NODE-WALKS WITH $\mathbb{R}^d$-WALKS

Original GRF-method relies on the random walks on the nodes of the input graph. SWING relaxes this procedure with walks conducted in the $d$-dimensional space where i-graphs of interest are embedded. Each walk starts in the graph node (identified with its corresponding feature vector), as for GRFs.

Let $\mathbf{x}^t(\mathbf{p}) \in \mathbb{R}^d$ be the location of the walker, that originates its walk in $\mathbf{p}$, after $t$ steps for $t = 0, 1, ..., T$. We start with this rule of the $t \to t + 1$ transition, where $\{\tau_i^t\}_{i=1,...,N}^{t=0,...,T}$ are chosen iid from Gumbel distribution.:

$$\mathbf{x}^{t+1}(\mathbf{p}) = \sum_{i=1}^N \frac{\exp(\log(f(\mathbf{p}_i - \mathbf{x}^t(\mathbf{p})) + \tau_i^t)\sigma^{-2})\mathbf{p}_i}{\sum_{j=1}^N \exp(\log(f(\mathbf{p}_j - \mathbf{x}^t(\mathbf{p})) + \tau_j^t)\sigma^{-2})}, \tag{3}$$

Note that the new location is defined as a convex sum of the feature vectors $\mathbf{p}_i$ with coefficients given by the probabilities defining Gumbel-softmax distribution with temperature $\sigma^2$. That convex sum is the differentiable relaxation of the

following update rule:

$$\mathbf{x}^{t+1}(\mathbf{p}) = \mathbf{p}_{\text{argmax}_{i=1}^N[\log(f(\mathbf{p}_i - \mathbf{x}^t(\mathbf{p})) + \tau_i^t)]}, \quad (4)$$

applying the so-called *Gumbel-trick* (Jang et al., 2017). It is well known that a discrete distribution on the set $\{1, ..., N\}$ given by the formula index $= \text{argmax}_{i=1}^N[\log(f(\mathbf{p}_i - \mathbf{x}^t(\mathbf{p})) + \tau_i^t)]$ is a categorical distribution with the corresponding probabilities $(p_1, ..., p_N)$ satisfying the following: $p_i \sim f(\mathbf{p}_i - \mathbf{x}^t(\mathbf{p}))$.

We conclude that Eq. 4 effectively describes a random walk on the nodes of the input graph with a probablity of transitioning from $i$ to $j$ proportional to the weight $w_{i,j} = f(v(i) - v(j))$. This is exactly GRF algorithm. Equation 3 constitutes a convenient relaxation, where random walks explore entire $\mathbb{R}^d$ rather than just a discrete space of graph's nodes (with the approximation of Eq. 4 more accurate for smaller temperatures $\sigma^2 > 0$). However it is not efficient to compute since it requires $O(N^2 Tm)$ computational time for $Nm$ conducted walks (where $m$ stands for the number of random walks per given starting vertex).

### 2.2.2. RANDOM FEATURES TO THE RESCUE

To obtain an algorithm with the linear dependence on $N$, we apply *random features* (RFs) (Rahimi & Recht, 2007; 2008). RFs is a method to linearize bi-variate functions $K : \mathbb{R}^d \times \mathbb{R}^d \to \mathbb{R}$ (often kernels) via randomized $\phi_1, \phi_2 : \mathbb{R}^d \to \mathbb{C}^r$ (usually $\phi_1 = \phi_2$), as: $K(\mathbf{x}, \mathbf{y}) \approx \phi_1^\top(\mathbf{x})\phi_2(\mathbf{y})$, where: $r$ is the number of RFs.

Define K as: $K(\mathbf{x}, \mathbf{y}) = (f(\mathbf{x} - \mathbf{y}))^{\frac{1}{\sigma^2}}$. Let $\phi_{f,\sigma}$ be a mapping providing its corresponding RFs, i.e. $(f(\mathbf{x} - \mathbf{y}))^{\frac{1}{\sigma^2}} = \mathbb{E}[\phi_{f,\sigma}^\top(\mathbf{x})\phi_{f,\sigma}(\mathbf{y})]$ (in Sec. 2.3 we will show how maps $\phi_{f,\sigma}$ can be constructed). We can then write an approximate version of the transition Eq. 3, as:

$$\mathbf{x}^{t+1}(\mathbf{p}) = \sum_{i=1}^N \frac{\mathbf{p}_i \cdot \phi_{f,\sigma}^\top(\mathbf{p}_i)\phi_{f,\sigma}(\mathbf{x}^t(\mathbf{p})) \exp(\frac{\tau_i^t}{\sigma^2})}{\sum_{j=1}^N \phi_{f,\sigma}^\top(\mathbf{p}_j)\phi_{f,\sigma}(\mathbf{x}^t(\mathbf{p})) \exp(\frac{\tau_j^t}{\sigma^2})} =$$

$$\frac{\left(\sum_{j=1}^N \mathbf{p}_j \exp(\frac{\tau_j^t}{\sigma^2})\phi_{f,\sigma}^\top(\mathbf{p}_j)\right)\phi_{f,\sigma}(\mathbf{x}^t(\mathbf{p}))}{\left(\sum_{j=1}^N \exp(\frac{\tau_j^t}{\sigma^2})\phi_{f,\sigma}^\top(\mathbf{p}_j)\right)\phi_{f,\sigma}(\mathbf{x}^t(\mathbf{p}))} \quad (5)$$

**Exponentiated Gumbel distribution factorization:** We will make one more refinement, to further improve the quality of the approximation. Denote by $\mathcal{F}_{\sigma^2}$ the distribution of $\exp(\frac{\tau^2}{\sigma^2})$, where $\tau$ is Gumbel. Let $P_a$ and $P_b$ be two distributions with probability density functions:

$$p_a(y) = \sigma^2 y^{-1-2\sigma^2} \exp\left(-\frac{1}{2y^{2\sigma^2}}\right), \quad (6)$$

$$p_b(y) = \sqrt{\frac{2}{\pi}}\sigma^2 y^{-1-\sigma^2} \exp\left(-\frac{1}{2y^{2\sigma^2}}\right). \quad (7)$$

We have (proof in Appendix A, see: Proposition A.1):

**Proposition 2.1.** *If $a \sim P_a$ and $b \sim P_b$ are independent, then $a \cdot b \sim \mathcal{F}_{\sigma^2}$.*

Thus we will replace variables $\exp(\frac{\tau_i^t}{\sigma^2})$ in Eq. 5 with the products: $\alpha(\mathbf{p}_i)\beta(\mathbf{x}^t(\mathbf{p}))$, with $\alpha(*), \beta(*)$ sampled iid from the distributions with density functions $p_a$ and $p_b$ respectively. We can now summarize the walk step.

Define: $A_{f,\sigma} = \sum_{j=1}^N \mathbf{p}_j\alpha(\mathbf{p}_j)\phi_{f,\sigma}^\top(\mathbf{p}_j) \in \mathbb{R}^{d \times r}$ and $B_{f,\sigma} = \sum_{j=1}^N \alpha(\mathbf{p}_j)\phi_{f,\sigma}^\top(\mathbf{p}_j) \in \mathbb{R}^r$. Note that $A_{f,\sigma}$ and $B_{f,\sigma}$ are of size **independent** from $N$. Furthermore, they do not depend on the random walks conducted. We conclude that both can be pre-computed in time linear in $N$ and then used in the computation of $\mathbf{x}^{t+1}(\mathbf{p})$, as follows: $\mathbf{x}^{t+1}(\mathbf{p}) = \frac{A_{f,\sigma}\phi_{f,\sigma}(\mathbf{x}^t(\mathbf{p}))\beta(\mathbf{x}^t(\mathbf{p}))}{B_{f,\sigma}\phi_{f,\sigma}(\mathbf{x}^t(\mathbf{p}))\beta(\mathbf{x}^t(\mathbf{p}))}$. This can be done in time $O_N(1)$. Thus the computation of all walks can be done in time $O(NTm)$, linear in $N$.

### 2.2.3. RELAXED LOAD COMPUTATION

The next step towards SWING algorithm is the relaxation of the load update rule from Algorithm 1:

$$\text{load} \leftarrow \text{load} \times \frac{\deg(i)}{1 - p_{\text{halt}}}, \quad (8)$$

where $i$ is the current node, $j$ is the next node, $\mathbf{W}[i, j] = w_{i,j} = f(v(i) - v(j))$ and $\deg(i) = \sum_{j=1}^N w_{i,j}$.

As in the previous section, we will relax this update rule, using mapping $\phi_{f,\sigma}$. Denote by $l^t(\mathbf{p})$ the load after $t$ steps for the walk originating in $p$. We obtain:

$$l^{t+1}(\mathbf{p}) = \frac{1}{1 - p_{\text{halt}}}l^t(\mathbf{p})\left(\sum_{i=1}^N \phi_{f,\sigma}^\top(\mathbf{p}_i)\right)\phi_{f,\sigma}(\mathbf{x}^t(\mathbf{p})) \quad (9)$$

Thus the load update can be conducted in time $O_N(1)$, at the cost of the one-time precomputation of the vector $C_{f,\sigma} = \sum_{i=1}^N \phi_{f,\sigma}^\top(\mathbf{p}_i) \in \mathbb{R}^r$, of linear in $N$ time complexity. Thus all the load updates throughout the execution of all the walks can be done in time $O(NTm)$.

### 2.2.4. THE ACTIONS OF THE KERNEL MATRIX

Let us denote: $u^t(\mathbf{p}) = l^t(\mathbf{p})\rho(t)$, where $\rho$ is the modulation function from Algorithm 1. Scalar $l^t(\mathbf{p})$ should be thought of as a relaxed version of the load after $t$ steps for the walk that originates in the vertex corresponding to vector $\mathbf{p}$. Thus $u^t(\mathbf{p})$ is a relaxed version of the additive term from l.5 of Algorithm 1, describing how the value of the entry of the signature vector $\xi_\rho$, corresponding to vertex $j$ that the walk visits, changes.

In our relaxed version, no vertex is visited, but instead the walk evolves between the vertices. Thus, instead of

depositing $u^t(\mathbf{p})$ entirely in one of the vertices, we will instead "spread it" across all the vertices. The amount deposited in vertex $k$ corresponding to $\mathbf{p}_k$ will be proportional to $g(\mathbf{x}^t(\mathbf{p}) - \mathbf{p}_k)$ for some function: $g : \mathbb{R}^d \to \mathbb{R}$. As before, rather than working directly with $g$, we will instead work with the linearized versions of the function $\mathrm{J}(\mathbf{x}, \mathbf{y}) = (g(\mathbf{x} - \mathbf{y}))^{\frac{1}{\sigma^2}}$, with the corresponding RF-map $\psi_g$. That means that our signature vectors $\xi_\rho$ are dense, as opposed to sparse $\xi_\rho$ for the original GRFs, a clear computational challenge.

Instead of explicitly storing and updating $N$ dense $N$-dimensional signature vectors, requiring $\Omega(N^2)$ space and time complexity, we will apply them implicitly, showing an efficient algorithm for computing the action of the kernel matrix $\mathbf{K} = [\xi_\rho^\top(i)\xi_\rho(j)]_{i,j=1,...,N} \in \mathbb{R}^{N \times N}$ on a given vector $\mathbf{v} \in \mathbb{R}^N$. As in the case of regular GRFs (see: Sec. 2.1), we will leverage the decomposition: $\mathbf{K} = \mathbf{K}_1\mathbf{K}_2^\top$, given by signature vectors. We will show a linear time complexity algorithm (in $N$) for computing $\mathbf{K}_1\mathbf{v}$ and $\mathbf{K}_2^\top\mathbf{v}$ for any $\mathbf{v} \in \mathbb{R}^N$.

It suffices to show that the following two expressions: $\sum_{k=1}^{N} \xi_\rho(i)[k]v_k$ (for a fixed $i \in \{1, ..., N\}$) and $\sum_{i=1}^{N} \xi_\rho(i)[k]v_k$ (for a fixed $k \in \{1, ..., N\}$) can be computed in $O(1)$ time. The following holds:

$$
\sum_{k=1}^{N} \xi_\rho(i)[k]v_k = \sum_{t=1}^{T}\sum_{k=1}^{N} u^t(\mathbf{p}_i) \frac{\psi_g^\top(\mathbf{x}^t(\mathbf{p}_i))\psi_g(\mathbf{p}_k)v_k}{\sum_{j=1}^{N} \psi_g^\top(\mathbf{x}^t(\mathbf{p}_i))\psi_g(\mathbf{p}_j)}
$$
$$
= \sum_{t=1}^{T} \frac{u^t(\mathbf{p}_i)\psi_g^\top(\mathbf{x}^t(\mathbf{p}_i))}{\psi_g^\top(\mathbf{x}^t(\mathbf{p}_i))\sum_{j=1}^{N}\psi_g(\mathbf{p}_j)} \sum_{k=1}^{N}\psi_g(\mathbf{p}_k)v_k
$$
$$
\sum_{i=1}^{N} \xi_\rho(i)[k]v_k = \sum_{t=1}^{T}\sum_{i=1}^{N} u^t(\mathbf{p}_i) \frac{\psi_g^\top(\mathbf{x}^t(\mathbf{p}_i))\psi_g(\mathbf{p}_k)v_k}{\sum_{j=1}^{N} \psi_g^\top(\mathbf{x}^t(\mathbf{p}_i))\psi_g(\mathbf{p}_j)}
$$
$$
= \sum_{t=1}^{T}\sum_{i=1}^{N} \left( u^t(\mathbf{p}_i) \frac{\psi_g^\top(\mathbf{x}^t(\mathbf{p}_i))}{\psi_g^\top(\mathbf{x}^t(\mathbf{p}_i))\sum_{j=1}^{N}\psi_g(\mathbf{p}_j)} \right) \psi_g(\mathbf{p}_k)v_k
$$
$$(10)$$

We see that both expressions can be clearly calculated in time $O(NT)$, at the cost of the one-time pre-computations of linear in $N$ time complexity. To be more specific, for the former expresion: $\sum_{k=1}^{N} \xi_\rho(i)[k]v_k$, one needs to pre-compute $\sum_{k=1}^{N} \psi_g(\mathbf{p}_k)v_k$ and $C = \sum_{k=1}^{N} \psi_g(\mathbf{p}_k)$, both in time linear in $N$. Then, for any given $i$, one needs to compute $\sum_{t=1}^{T} \frac{u^t(\mathbf{p}_i)\psi_g^\top(\mathbf{x}^t(\mathbf{p}_i))}{\psi_g^\top(\mathbf{x}^t(\mathbf{p}_i))C}$, in time $O_N(1)$. On the other hand, for the latter expression, one needs to pre-compute $D = \sum_{t=1}^{T}\sum_{i=1}^{N} \left( u^t(\mathbf{p}_i) \frac{\psi_g^\top(\mathbf{x}^t(\mathbf{p}_i))}{\psi_g^\top(\mathbf{x}^t(\mathbf{p}_i))C} \right)$ in time $O(NT)$. Then, for any given $k$, it suffices to compute $D\psi_g(\mathbf{p}_k)v_k$, in time $O_N(1)$.

Thus, by running $m$ random walks per node, as in Sec. 2.1 and averaging over $m$ obtained signature vectors (as in l.13

of Algorithm 1), we obtain SWING algorithm for computing the action of the kernel matrix on an arbitrary $\mathbf{v} \in \mathbb{R}^N$ of time complexity $O(NTm)$.

## 2.3. How to compute maps $\phi_f$ and $\psi_g$ ?

Note that in order to construct $\phi_f, \psi_g : \mathbb{R}^d \to \mathbb{C}^m$, one needs to be able to re-write function $h$ ($h \in \{f, g\}$) as:

$$
h(\mathbf{x} - \mathbf{y}) \approx \eta_1^\top(\mathbf{x})\eta_2(\mathbf{y}) \tag{11}
$$

for some randomized $\eta_1, \eta_2$ (effectively playing a role of $\phi$ or $\psi$). In principle, we would like $\eta_1 = \eta_2$, but it is easy to see that for the asymmetric setting $\eta_1 \neq \eta_2$ our previous analysis remains valid, with only cosmetic changes needed. Thus, without loss of generality, we will consider two: $\eta_1$, $\eta_2$. We will show a systematic method to construcy $\eta_1, \eta_2$. Note that we have:

$$
h(\mathbf{z}) = \int_{\mathbb{R}^d} \exp(-2\pi k\omega^\top\mathbf{z})\tau(\omega)d\omega, \tag{12}
$$

where $\tau$ is the inverse Fourier Transform of $h$:

$$
\tau(\omega) = \int_{\mathbb{R}^d} \exp(2\pi k\mathbf{x}^\top\omega)h(\mathbf{x})d\mathbf{x} \tag{13}
$$

($k^2 = -1$). Thus, taking: $\mathbf{z} = \mathbf{x} - \mathbf{y}$, we can rewrite:

$$
h(\mathbf{x} - \mathbf{y}) = C \cdot \mathbb{E}_{p(\omega)}[\exp(-2\pi k\omega^\top\mathbf{x})\exp(2\pi k\omega^\top\mathbf{y})], \tag{14}
$$

where $p(\omega)$ is the probabilistic distribution with density proportional to $\tau(\omega)$ and $C = \int_{\mathbb{R}^d} \tau(\omega)d\omega$ (here we assume that the latter integral is well-defined). Thus, if we can efficiently (potentially approximately) sample from $p(\omega)$, then we can leverage the following:

$$
h(\mathbf{x} - \mathbf{y}) \stackrel{\mathbb{E}}{=} \eta_1^\top(\mathbf{x})(\eta_2(\mathbf{y})), \tag{15}
$$

where for $\omega_1, ..., \omega_m \sim p(\omega)$ and for some $r \in \mathbb{N}$ (number of random features) the following holds:

$$
\eta_1(\mathbf{v}) = \sqrt{\frac{C}{r}} \left( \exp(-2\pi k\omega_1^\top\mathbf{v}), ..., \exp(-2\pi k\omega_r^\top\mathbf{v}) \right),
$$
$$
\eta_2(\mathbf{v}) = \sqrt{\frac{C}{r}} \left( \exp(2\pi k\omega_1^\top\mathbf{v}), ..., \exp(2\pi k\omega_r^\top\mathbf{v}) \right). \tag{16}
$$

### 2.3.1. RADIAL BASIS FUNCTIONS

The above technique applies also for a prominent class of radial basis (Gaussian kernel) functions $h$ defined as: $h(\mathbf{z}) = \exp(-\frac{\|\mathbf{z}\|_2^2}{2\sigma^2})$ for some $\sigma > 0$. For them there exist particularly efficient ways of constructing $\eta$-maps, in particular the so-called *positive random features* (Choromanski

et al., 2021). For these functions $h$, we have the following, where $\omega_1, ..., \omega_r \sim \mathcal{N}(0, \mathbf{I}_d)$:

$$h(\mathbf{x} - \mathbf{y}) \stackrel{\mathbb{E}}{=} \eta_+(\mathbf{x})^\top \eta_+(\mathbf{y}),$$

$$\eta_+(\mathbf{u}) \stackrel{\text{def}}{=} \frac{1}{\sqrt{r}} \exp\left(-\frac{\|\mathbf{u}\|_2^2}{\sigma^2}\right) \left(\exp(\omega_1^\top \frac{\mathbf{u}}{\sigma}), ..., \exp(\omega_r^\top \frac{\mathbf{u}}{\sigma})\right)$$

(17)

Additional techniques can be applied to reduce the variance of that $\eta$-induced approximation of $h(\mathbf{x} - \mathbf{y})$. A standard approach is to apply block-orthogonal or simplex ensembles $\{\omega_1, ..., \omega_r\}$ rather than consisting of iid sampled Gaussian vectors (Yu et al., 2016; Reid et al., 2023). Another option is importance sampling, which we found to be particularly effective. Further details are provided in Appendix B.

### 2.3.2. STEP FUNCTIONS AND $\epsilon$-NEIGHBORHOODS

Let us get back to the step functions considered in the introduction (Sec. 1) and given as: $h(\mathbf{z}) = \mathbb{I}(\|\mathbf{z}\| \leq \epsilon)$. If $f$ takes this form, then we effectively obtain an $\epsilon$-neighborhood graph. Depending on the definition of the norm $\|\|$, we obtain different formulae on the inverse Fourier Transform $\tau$, but interestingly, often of the compact form (to apply our previous Fourier analysis). For instance, for the $L_1$-norm, we have:

$$\tau(\mathbf{v}) = \prod_{i=1}^{d} \frac{\sin(2\epsilon v_i)}{v_i}.$$

(18)

On the other hand, for the $L_2$-norm $\|\|_2$, $\tau$ becomes the so-called d-th order *Bessel function* (Grafakos, 2010).

### 2.4. Putting this altogether

SWING algorithm provides an efficient way of computing an approximate action of the graph kernel matrix on a given vector. As in the regular GRF setting, it applies random walks, but in contrast to regular GRFs, those walks evolve in $\mathbb{R}^d$ rather than on the nodes of the graph. An efficient implementation of these walks is provided in Sec. 2.2.1. As those walks evolve, SWING updates the load scalar. An efficient implementation of those updates is provided in Sec. 2.2.3. The final algorithm, leveraging those loads across all steps of the walk is provided in Sec. 2.2.4. Algorithm 2 summarizes the resulting SWING procedure.

For simplicity, we assume that all the walks have the same length $T$ that is sampled from the Bernoulli distribution with failure probability $p_{\text{halt}}$. Since $0 < p_{\text{halt}} < 1$ is a graph-independent constant, an average walk length $\mathbb{E}[T] = \frac{1 - p_{\text{halt}}}{p_{\text{halt}}}$ is also a constant.

### 2.5. Scope and Approximation Properties

We conclude this section by discussing the scope of SWING and the approximation guarantees that motivate its use for

---

**Algorithm 2 SWING:** Approximate the action $\mathbf{K}_{\boldsymbol{\alpha}}(\mathbf{W})\mathbf{V}$ of the graph kernel on attribute matrix $\mathbf{V}$

**Input:** point cloud $\mathbf{p} \in \mathbb{R}^{N \times d}$, attribute matrix $\mathbf{V} \in \mathbb{R}^{N \times d_v}$, modulation function $\rho$, RF maps $\phi_{f,\sigma}$ and $\psi_g$, halting probability $p_{\text{halt}} \in (0, 1)$, number of walks $m$.

**Output:** approximation $\widehat{\mathbf{Y}} \approx \mathbf{K}_{\boldsymbol{\alpha}}(\mathbf{W})\mathbf{V}$.

1: initialize: $\widehat{\mathbf{Y}} \leftarrow \mathbf{0} \in \mathbb{R}^{N \times d_v}$
2: Sample $m$ walk lenghts: $\mathbf{T} \sim 1 + \text{Geom}(p_{\text{halt}})$
3: **for** $n_{\text{steps}} \in \mathbf{T}$ **do**
4:     sample $\omega$ and $\omega'$ for RF maps $\phi_{f,\sigma}^{(\omega)}$ and $\psi_g^{(\omega')}$
5:     $\boldsymbol{\Phi}_{\mathbf{p}} \leftarrow \phi_{f,\sigma}^{(\omega)}(\mathbf{p})$         ▷ precompute evolution RFs
6:     $\boldsymbol{\Psi}_{\mathbf{p}} \leftarrow \psi_g^{(\omega')}(\mathbf{p})$         ▷ precompute load RFs
7:     $\mathbf{L} \leftarrow \boldsymbol{\Psi}_{\mathbf{p}}^\top \mathbf{V}$         ▷ precompute KV for load
8:     $\mathbf{X}^0 \leftarrow \mathbf{p}$         ▷ initialize walkers
9:     $\widehat{\mathbf{Y}}_{\text{walk}} \leftarrow \mathbf{V} \cdot \rho(0)$         ▷ initialize approximation
10:     **for** $t = 1, \ldots, n_{\text{steps}} - 1$ **do**
11:         $l_t \leftarrow \frac{\rho(t)}{(1 - p_{\text{halt}})^t}$         ▷ Compute load
12:         $\boldsymbol{\Phi}_{\mathbf{X}} \leftarrow \phi_{f,\sigma}^{(\omega)}(\mathbf{X^{t-1}})$         ▷ RFs for evolution
13:         $\mathbf{X}^t \leftarrow \text{Evolve}(\boldsymbol{\Phi}_{\mathbf{X}}, \boldsymbol{\Phi}_{\mathbf{p}}, \mathbf{p})$     ▷ sample Gumbel, normalize and evolve
14:         $\boldsymbol{\Psi}_{\mathbf{X}} \leftarrow \psi_g^{(\omega')}(\mathbf{X^t})$         ▷ RFs for load distribution
15:         $\boldsymbol{\Psi}_{\mathbf{X}} \leftarrow \text{Normalize}(\boldsymbol{\Psi}_{\mathbf{X}}, \boldsymbol{\Psi}_{\mathbf{p}})$
16:         $\widehat{\mathbf{Y}}_{\text{walk}} \leftarrow \widehat{\mathbf{Y}}_{\text{walk}} + l_t \boldsymbol{\Psi}_{\mathbf{X}} \mathbf{L}$     ▷ distribute load
17:     **end for**
18:     $\widehat{\mathbf{Y}} \leftarrow \widehat{\mathbf{Y}} + \widehat{\mathbf{Y}}_{\text{walk}}$
19: **end for**
20: $\widehat{\mathbf{Y}} \leftarrow \widehat{\mathbf{Y}}/m$         ▷ normalize

---

implicit graphs.

SWING is intended for implicitly defined graphs, where edge weights are specified by a function of node features rather than by an explicitly materialized adjacency matrix. This setting covers common point-cloud constructions, including Gaussian-affinity and $k$-nearest-neighbor graphs, and allows the method to operate directly on dense tensors of node features with memory linear in the number of nodes.

The approximation error admits a standard random-feature analysis, following Claim 1 in Rahimi & Recht (2007). For bounded positive random features, the probability of a uniform kernel-approximation error larger than $\epsilon$ over all queried pairs can be bounded by an expression proportional to $\exp\left(-\frac{r}{d}\right) \left(\frac{\text{diam}(\mathcal{M})}{\epsilon}\right)^2$, where $r$ is the number of random features, $d$ is the ambient dimension, and $\text{diam}(\mathcal{M})$ is the diameter of the domain containing the points. In particular, the number of random features required for a fixed accuracy scales with the feature dimension and domain size, but not directly with the number of graph nodes.

SWING also introduces approximation error through the

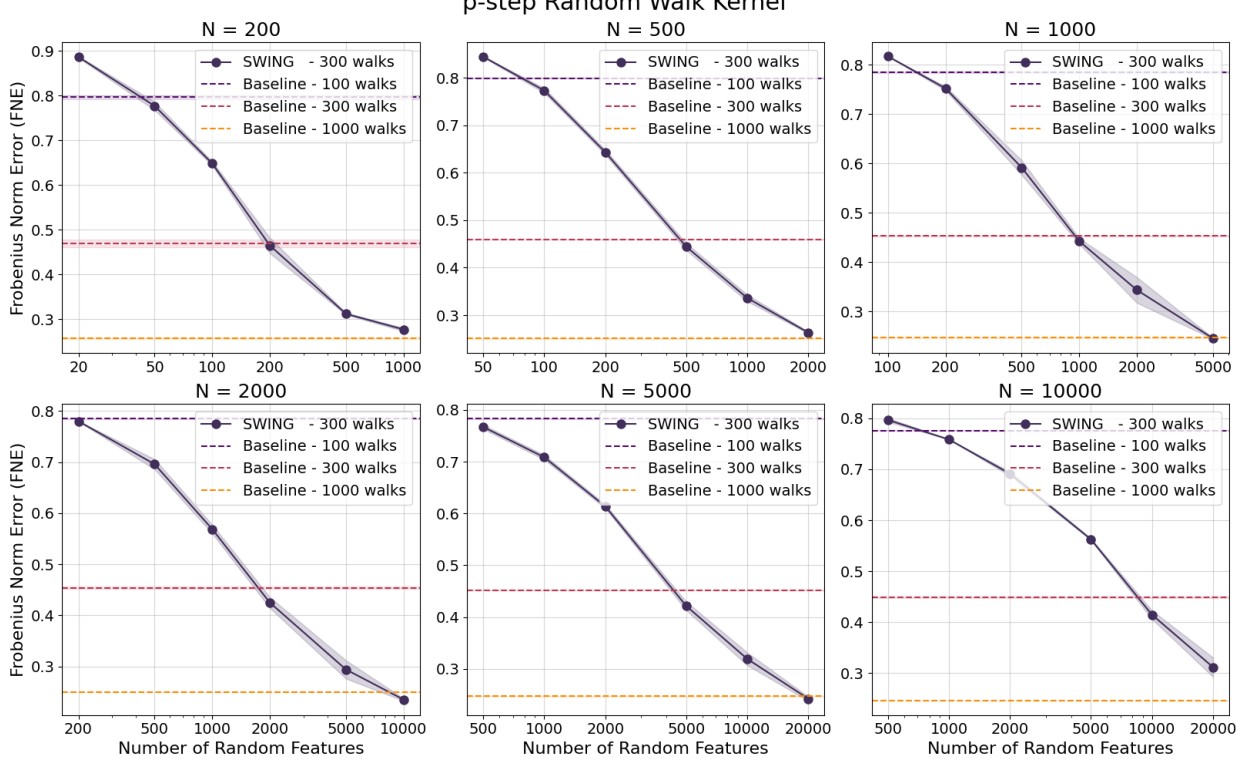

*Figure 2.* Frobenius norm error on synthetic point clouds of increasing size for the $p$-step random walk kernel.

Gumbel–Softmax relaxation and through accumulation along walks. Empirically, however, the former does not degrade performance; rather, as shown in Figures 2, 7, and 8, there exists a threshold in the number of random features $r$ beyond which SWING outperforms GRFs without Gumbel–Softmax relaxation. We hypothesize that the continuous relaxation and load distribution allow SWING to explore multiple walks simultaneously at the cost of a single relaxed walk. For the latter source of error, $T$ is chosen as a small constant, as in prior GRF methods, making the resulting accuracy–efficiency trade-off effective for the implicit-graph regime considered here.

## 3. Experiments

We evaluate the proposed method through a set of controlled synthetic and real-world experiments designed to assess both approximation accuracy and computational efficiency.

In Section 3.1, we empirically show that SWING accurately approximates multiple graph kernels on synthetic 3D point clouds of increasing size, and that it consistently traces the Pareto-optimal error-time frontier when compared to standard GRF methods (Choromanski, 2023; Reid et al., 2024).

We then complement these results in Section 3.2 with experiments on vertex normal prediction for 3D meshes from

the Thingi10K dataset (Zhou & Jacobson, 2016) and image classification using Vision Transformers on ImageNet (Deng et al., 2009) and Places365 (Zhou et al., 2018). These results demonstrate that the proposed approach achieves performance comparable to or better than existing methods, while substantially reducing computational cost.

### 3.1. Synthetic PCs experiment

#### 3.1.1. EMPIRICAL KERNEL APPROXIMATION

We first verify that the proposed method provides an unbiased estimate of several graph kernels on a set of synthetic 3D point clouds. In this setting, the ground-truth weight matrix $\mathbf{W}$ is constructed using a Gaussian kernel applied to the point coordinates. Following the notation in Equation (1), the weights are defined as

$$w_{i,j} = \exp\left(-\frac{\|\mathbf{x}_i - \mathbf{x}_j\|^2}{2}\right). \tag{19}$$

Let $N$ denote the number of points in the point cloud. Each point coordinate is sampled independently from the uniform distribution

$$\mathcal{U}\left[-\sqrt[3]{\frac{\pi N}{6}}, \sqrt[3]{\frac{\pi N}{6}}\right]^3.$$

This choice ensures a constant expected point density as

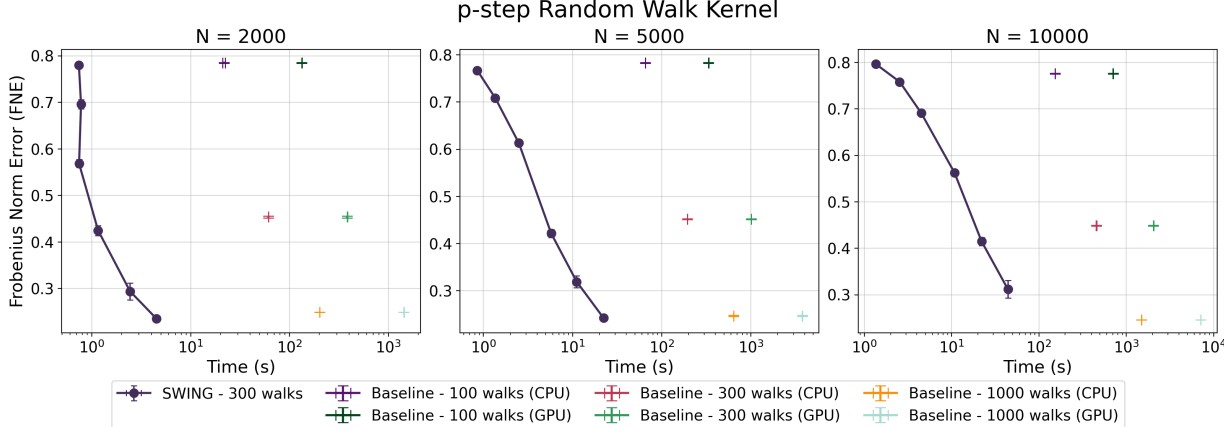

*Figure 3.* Trade-off between approximation error (FNE) and computation time for the $p$-step random walk kernel on synthetic point clouds of varying size.

$N$ increases, so that, up to boundary effects, a ball of unit radius centered at a point contains approximately one other point on average.[1]

We evaluate SWING on point clouds with varying numbers of points $N \in \{200, 500, 1.000, 2.000, 5.000, 10.000\}$. Across all experiments, we set $p_{\text{halt}} = 0.3$ and use 300 random walks and use the gaussian kernel with positive random features both for the walk evolution part (Section 2.2.1) and the relaxed load distribution part (Section 2.2.3). Graph kernel approximation quality is measured using the relative Frobenius Norm Error (FNE) between the true kernel $\mathbf{K}$ and its approximation $\hat{\mathbf{K}}$, defined as

$$\text{FNE} = \frac{\|\mathbf{K} - \hat{\mathbf{K}}\|_F}{\|\mathbf{K}\|_F}.$$

We compare SWING with the standard GRFs approach (Choromanski, 2023; Reid et al., 2024). Figure 2 reports the kernel approximation error as a function of the number of random features for the $p$-step random walk kernel. Results for the diffusion kernel and the regularized laplacina kernel can be found in Appendix C in Figures 7 and 8 respectively.

The approximation quality improves monotonically as the number of random features increases. When both methods use the same number of random walks (i.e., 300), there is always a regime in which SWING achieves lower error than the baseline; this behavior is also observed for the other kernels reported in Appendix C.

### 3.1.2. TIME SPEEDUP OF SWING

Having established that the proposed method can accurately approximate different graph kernels, we now evaluate the

computational speedup it provides. Using the same experimental setting as before, we report the FNE-time trade-off for SWING and for the standard GRF approach, both in its original CPU implementation and in a modified version enabling GPU computation.

Figure 3 shows that SWING traces the Pareto-optimal frontier: by varying the number of random features, the accuracy-time trade-off can be smoothly adjusted while remaining on the Pareto front. The improvement is substantial, with inference speedups exceeding one order of magnitude. The same behavior is observed for the other graph kernels reported in Appendix C (Figures 9 and 10).

### 3.2. Downstream Experiments

In this subsections, we showcase the utility of SWING on various downstream tasks, including vertex normal prediction and image classification.

**Vertex Normal Prediction** We evaluate SWING on the task of normal vector prediction via mesh interpolation. Each vertex in a mesh $G(V, E)$ is associated with spatial coordinates $\mathbf{x}_i \in \mathbb{R}^3$ and a unit normal vector $\mathbf{F}_i \in \mathbb{R}^3$. Following Choromanski et al. (2023), we randomly sample a subset $V' \subset V$, where $|V|' = 0.8|V|$ and mask the corresponding vertex normals. The objective is to predict the normals of masked vertices $i \in V'$ according to $\mathbf{F}_i = \sum_{j \in V \setminus V'} \mathbf{K}(i, j) \mathbf{F}_j$, where $\mathbf{K}$ is the diffusion kernel. We report the average cosine similarity between predicted and ground-truth vertex normals across all vertices. SWING is evaluated on **40** meshes of 3D-printed objects of varying sizes from the Thingi10K dataset (Zhou & Jacobson, 2016). Additional experimental details are provided in Appendix D.1. Results for all meshes are reported in Table 2 (Appendix D.1), with a subset of larger meshes shown in Figure 4. SWING achives performance comparable to standard

---

[1]Neglecting boundary effects, the mean density over the support is $\frac{3}{4\pi}$. Since a ball of unit radius has volume $\frac{4\pi}{3}$, the expected number of points in such a ball is approximately one.

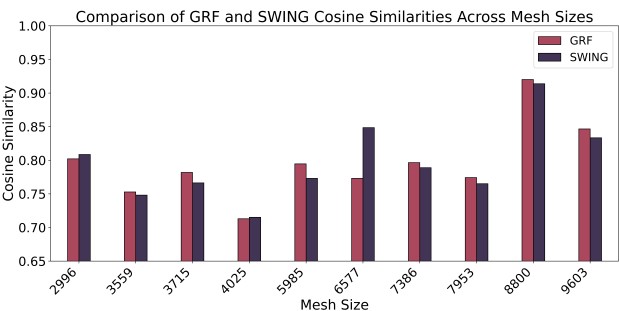

*Figure 4.* Cosine Similarities for some larger meshes. SWING performs at-par with GRFs.

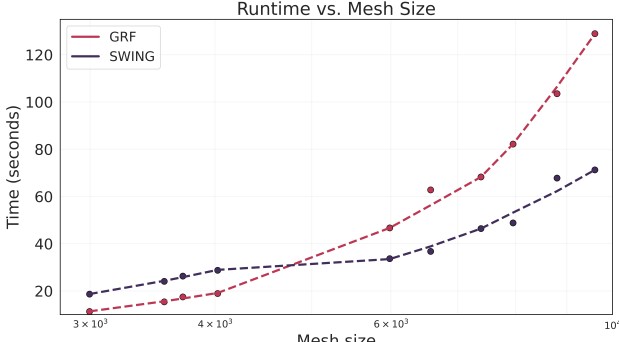

*Figure 5.* Kernel construction time on large meshes, measured on CPU. SWING demonstrates superior scalability compared to GRF.

GRFs while exhibiting improved computational efficiency, even when run on CPUs (Figure 5).

**ViT Experiments.** We evaluate the performance of SWING on the Vision Transformer (ViT; Dosovitskiy et al., 2020), where the image patches form an implicit grid graph with edges connecting spatial neighbors. Since the graph topology is static, the random walks required for SWING are pre-computed and frozen, introducing negligible overhead. This approximate kernel is subsequently normalized and utilized as a soft mask $\mathbf{M}$, which is added to the standard attention logic: $\text{Attention}(\mathbf{Q}, \mathbf{K}, \mathbf{V}) = \text{softmax}\left(\frac{\mathbf{Q}\mathbf{K}^\top}{\sqrt{d}} + \lambda\mathbf{M}\right)\mathbf{V}$.

We compare the standard ViT baseline against two GRFs-enhanced model on ImageNet (Deng et al., 2009) and Places365 (Zhou et al., 2018). The first one uses standard GRFs mechanism (ViT-GRFs) while the second uses our approximation (ViT-SWING). As shown in Table 1, ViT-SWING consistently improves performance. These results demonstrate that incorporating implicit graph topological information via SWING effectively boosts the predictive capability of the Vision Transformer model.

*Table 1.* Top-1 accuracy on ImageNet and Places365 validation sets. We compare the standard Vision Transformer (ViT) against two GRFs-enhanced variants. the SWING based one consistently outperforms the baseline across both datasets without requiring explicit graph materialization.

| Model | ImageNet | Places365 |
|---|---|---|
| ViT | 80.10 | 55.78 |
| ViT-GRFs | 80.09 | 55.39 |
| ViT-SWING (ours) | **80.34** | **55.97** |

## 4. Conclusion

We proposed in this paper SWING algorithms: Space Walks for Implicit Network Graphs. SWING is a relaxation of the prominent class of Graph Random Features (GRFs) methods, trading discrete walks on graph's nodes for walks in the $d$-dimensional embedding space of the graph's nodes, to obtain computational efficiency. SWING leverages several techniques such as: Gumbel softmax distributions, Fourier analysis and random features to achieve good approximation of the original combinatorial GRF algorithm. Our empirical evaluation confirms presented theoretical analysis, showing that significant computational speedups and no loss of accuracy can be obtained with SWING.

## Impact Statement

This paper presents work whose goal is to advance the field of Machine Learning. There are many potential societal consequences of our work, none which we feel must be specifically highlighted here. Since SWING completely bypasses all strictly combinatorial computations (e.g. random walks on graphs' nodes) via attention-based relaxations, it should lead to even wider adoption of the general class of GRF methods in downstream Machine Learning applications (heavily leveraging modern accelerators).

## Acknowledgments

AM and CA were supported by the Swiss National Science Foundation project FNS 204061, *HORD GNN: Higher-Order Relations and Dynamics in Graph Neural Networks*.

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

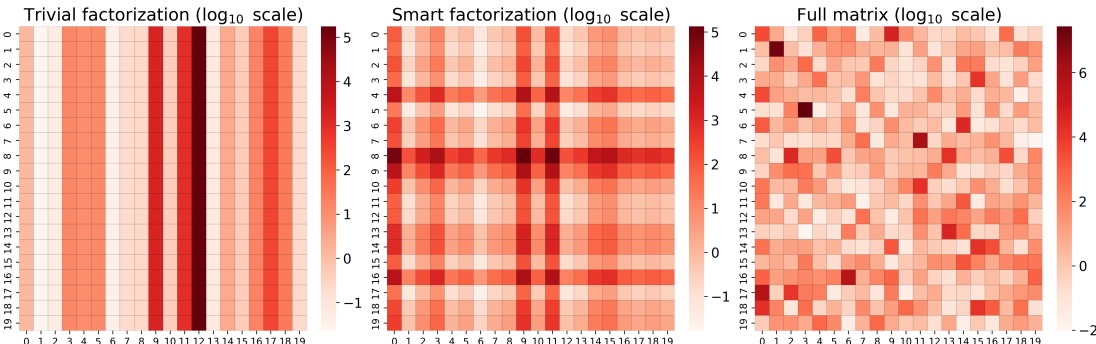

Figure 6. Comparison of matrix structures. Left: a trivial factorization concentrates mass along a few dominant directions. Center: the proposed factorization reveals structured interactions across factors. Right: the full matrix, shown for reference, where each entry is independently sampled requiring $\mathcal{O}(N^2)$ complexity.

# A. Factorizing Fréchet Noise

For each $\mathbf{p}$, each point $\mathbf{p}_i$ in the point cloud, and each time step $t$, a different sample from the Gumbel distribution should be drawn. To avoid $\mathcal{O}(N^2)$ complexity, we decompose $\exp(\frac{\tau_i^t}{\sigma^2})$ into factors that multiply the random features $\phi$.

For simplicity, we denote by $\phi_j$ the random features of the evolving trajectory and by $\phi_i$ the random features of each point in the point cloud, and we omit the time index $t$. We aim to decompose $\exp(\frac{\tau_{ij}}{\sigma^2})$ into two random factors $a_i \sim p_a$ and $b_j \sim p_b$ such that the product $a_i b_j$ has the same distribution as $\exp(\frac{\tau_{ij}}{\sigma^2})$. Note that if $\tau \sim$ Gumbel, then $\exp(\frac{\tau}{\sigma^2})$ follows a Fréchet distribution $\mathcal{F}_{\sigma^2}$ with scale parameter 1 and shape parameter $\sigma^2$.

The most trivial decomposition is $a_i \sim \mathcal{F}_{\sigma^2}$ and $b_j = 1$ (or vice versa). However, at each step, all points $\mathbf{p}$ would then be biased toward the same point cloud element corresponding to the largest Fréchet sample. Figure 6 (left) illustrates the reconstruction of $\exp(\frac{\tau_{ij}}{\sigma^2})$ under this trivial factorization.

To reduce this bias, we consider non-trivial distributions. The following proposition shows one such decomposition.

**Proposition A.1.** *Let $P_a$ and $P_b$ be two distributions with probability density functions*

$$p_a(y) = \sigma^2 y^{-1-2\sigma^2} \exp\left(-\frac{1}{2y^{2\sigma^2}}\right), \tag{20}$$

$$p_b(y) = \sqrt{\frac{2}{\pi}}\sigma^2 y^{-1-\sigma^2} \exp\left(-\frac{1}{2y^{2\sigma^2}}\right). \tag{21}$$

*If $a \sim P_a$ and $b \sim P_b$ are independent, then $a \cdot b \sim \mathcal{F}_{\sigma^2}$.*

*Proof.* A proof could be obtained directly using the product distribution formula for independent random variables (Feller, 1991). Instead, we provide a constructive proof based on the Mellin transform (Epstein, 1948).

The Mellin transform of a positive random variable $\xi$ with continuous pdf $f(y)$ is defined as

$$M_f(s) = \mathbb{E}_f\left[y^{s-1}\right] = \int_0^\infty y^{s-1} f(y)\, dy. \tag{22}$$

With a slight abuse of notation, we sometimes index $M$ by the distribution rather than by its pdf. The Mellin transform is particularly useful for analyzing products of random variables. Specifically, let $\xi_a$ and $\xi_b$ be independent, positive random variables with probability density functions $p_a$ and $p_b$ respectively. Assuming the relevant moments exist, the Mellin transform of the product $\xi_a \cdot \xi_b$ is $M_{p_a}(s)M_{p_b}(s)$.

Using the definition, one can easily verify that the Mellin transform of the Fréchet distribution $\mathcal{F}_{\sigma^2}$ is

$$M_{\mathcal{F}_{\sigma^2}}(s) = \Gamma\left(\frac{1-s}{\sigma^2} + 1\right). \tag{23}$$

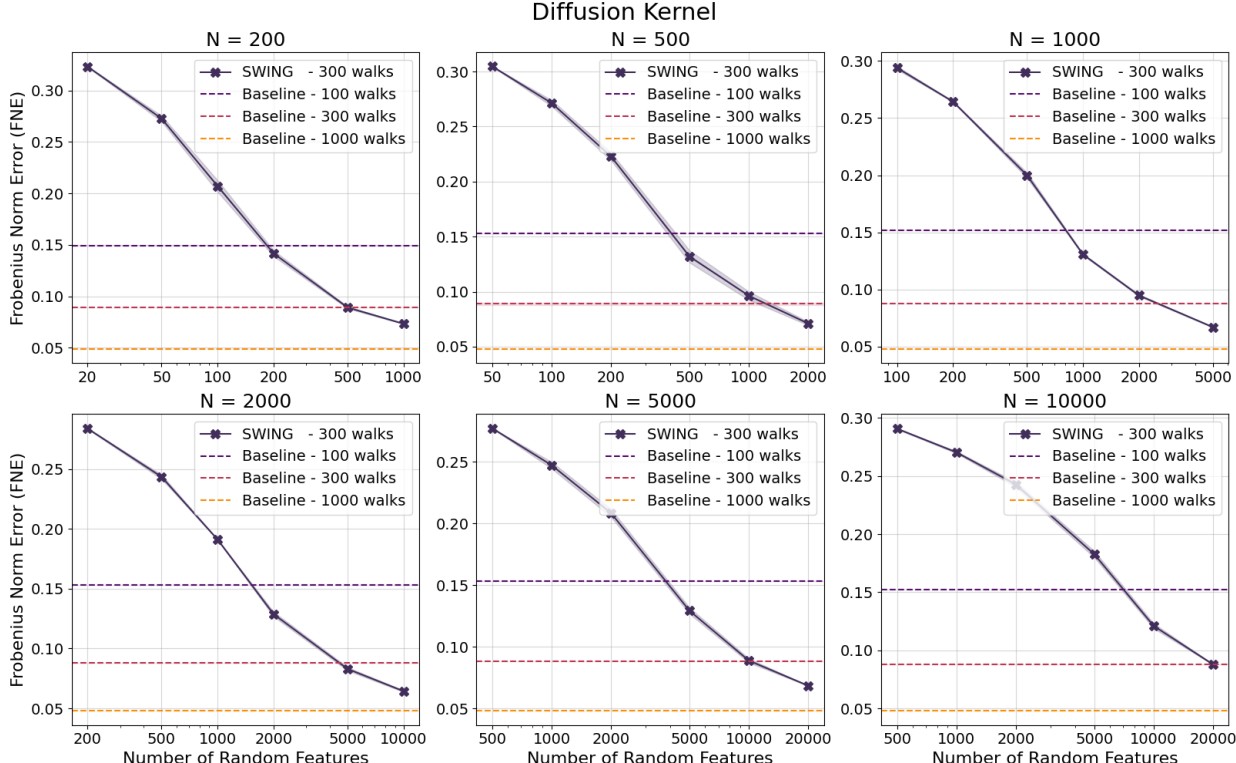

*Figure 7.* Frobenius norm error on synthetic point clouds of increasing size for the diffusion kernel.

Applying the duplication formula for the Gamma function, we can factor $M_{\mathcal{F}_{\sigma^2}}$ into two terms:

$$M_{\mathcal{F}_{\sigma^2}}(s) = \underbrace{2^{-\frac{s-1}{2\sigma^2}}\Gamma\left(\frac{1}{2\sigma^2} - \frac{s}{2\sigma^2} + 1\right)}_{M_{p_a}(s)} \cdot \underbrace{\frac{2^{-\frac{s-1}{2\sigma^2}}}{\sqrt{\pi}}\Gamma\left(\frac{1}{2\sigma^2} - \frac{s}{2\sigma^2} + \frac{1}{2}\right)}_{M_{p_b}(s)}. \tag{24}$$

We recall two basic properties of the Mellin transform. Let $\xi$ be a random variable with pdf $f$ and Mellin transform $M_f$:

M1) For $a > 0$, the random variable $a\xi$ has Mellin transform $a^{s-1}M_f(s)$.

M2) For $\alpha \in \mathbb{R}$, the random variable $\xi^{\alpha}$ has Mellin transform $M_f(\alpha s - \alpha + 1)$.

Using M1 and M2, it is straightforward to show that if $\phi_{\sigma^2} \sim \mathcal{F}_{\sigma^2}$, then the random variable $2^{-\frac{1}{2\sigma^2}}\phi_{\sigma^2}^{1/2}$ has Mellin transform $M_{p_a}$. Similarly, if $\eta$ follows the standard half-normal distribution, then $2^{-\frac{1}{2\sigma^2}}\eta^{-\frac{1}{\sigma^2}}$ has Mellin transform $M_{p_b}$. Applying the change-of-variables formula yields the densities in Equations 20 and 21.

$\square$

In Figure 6 (center) one can see how this factorization reduces the structure of the (implicit) random matrix.

## B. Importance Sampling for Gaussian Kernel

We observe that, when estimating the Gaussian kernel using positive random features, a systematic pattern emerges in the estimation quality. In particular, vectors with larger norms tend to yield kernel estimates that are systematically lower than the true values. This behavior can be explained intuitively through the following example.

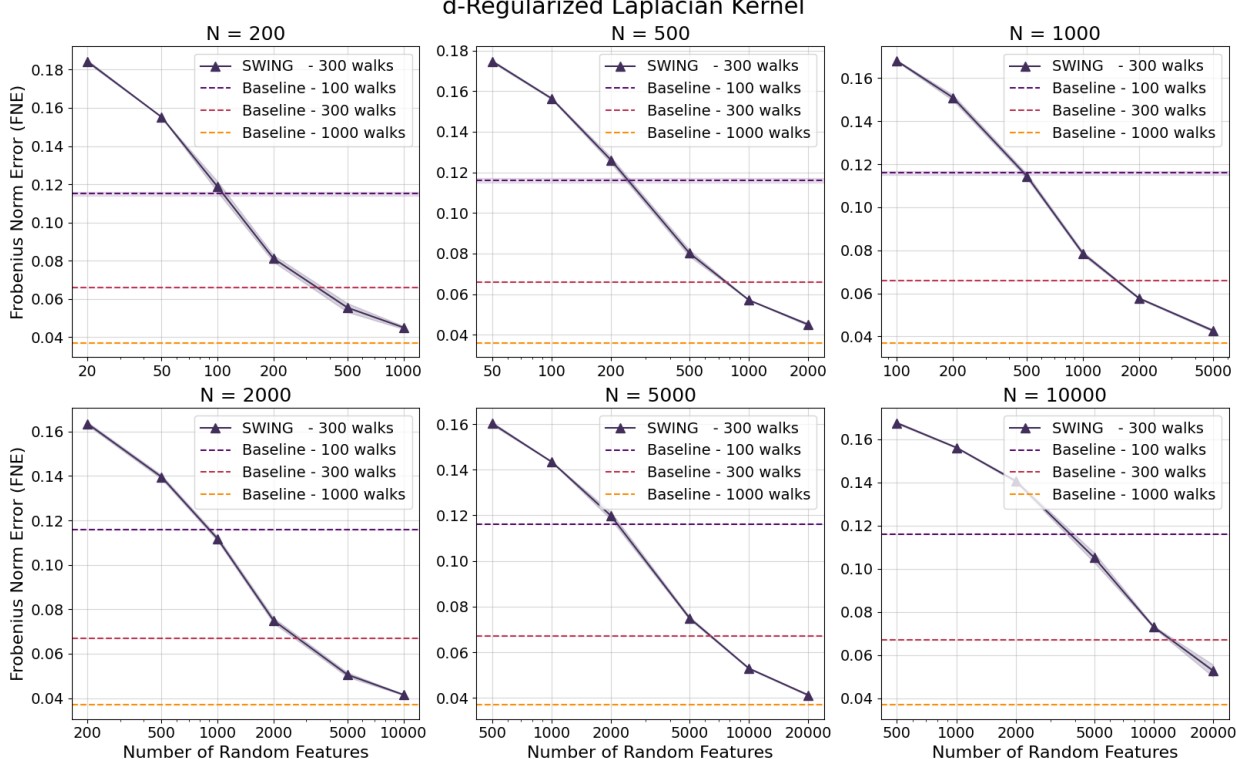

*Figure 8.* Frobenius norm error on synthetic point clouds of increasing size for the $d-$regularized laplacian kernel.

Consider one-dimensional vectors $x$ and $y$, the Gaussian kernel

$$K(x,y) = \exp\left(-\frac{(x-y)^2}{2}\right),$$

and the positive random feature

$$\eta_+(u) = \exp\left(-u^2 + \omega u\right).$$

The single-sample kernel estimate for $K(x,x)$ is

$$\hat{K}(x,x) = \exp\left(2x(\omega - x)\right).$$

For simplicity, assume $x > 0$ and $\omega > 0$. Since the true kernel value is $K(x,x) = 1$, accurate estimates occur when $\omega$ is sampled close to $x$. However, because $\omega \sim \mathcal{N}(0,1)$, such samples are extremely unlikely for points with large norms.

For this reason, we introduce importance sampling. Specifically, we propose, and find effective, sampling from the proposal distribution:

$$\omega_1,\ldots,\omega_r \sim \mathcal{N}\left(0, \alpha\,\mu, \mathbf{I}_d\right),$$

where $\mu$ denotes the median norm of the data points and $\alpha$ is a hyperparameter.

## C. Kernel Estimation on Synthetic Datasets for Additional Kernels

This appendix reports additional results on synthetic point clouds for the diffusion kernel and the $d$-regularized Laplacian kernel. All experiments follow the same experimental protocol described in Section 3: point clouds are generated with constant expected density, kernel weights are constructed using a Gaussian affinity, and performance is evaluated using the relative Frobenius Norm Error (FNE). The same values of $p_{\text{halt}}$, number of random walks, and random feature constructions are used throughout. For the hyperparameter search, we evaluated five Gumbel–Softmax temperatures uniformly spaced in the standard interval $[0.1, 1]$. For the scaling parameter, we evaluated five values logarithmically spaced in $[0.1, 10] \times N^{1/3}$.

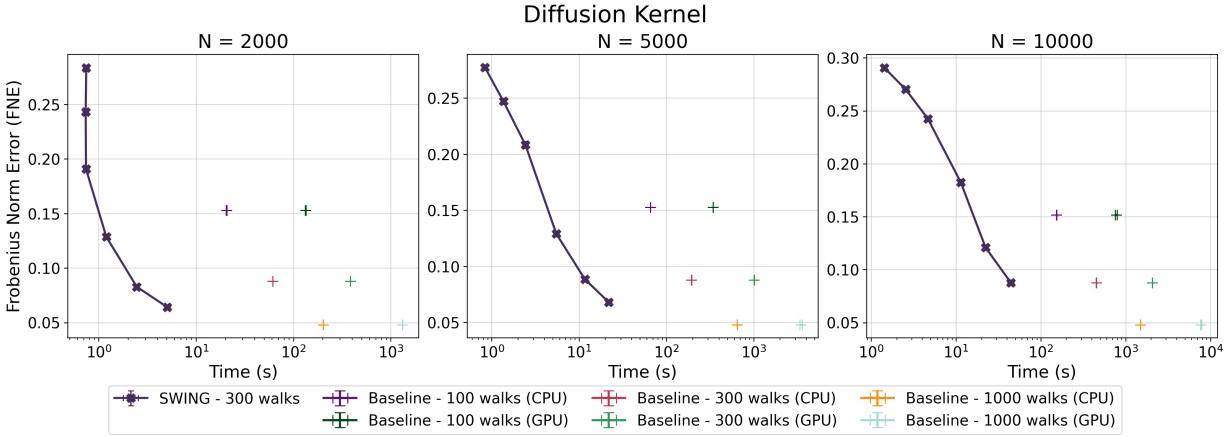

*Figure 9.* Trade-off between approximation error (FNE) and computation time for the diffusion kernel on synthetic point clouds of varying size.

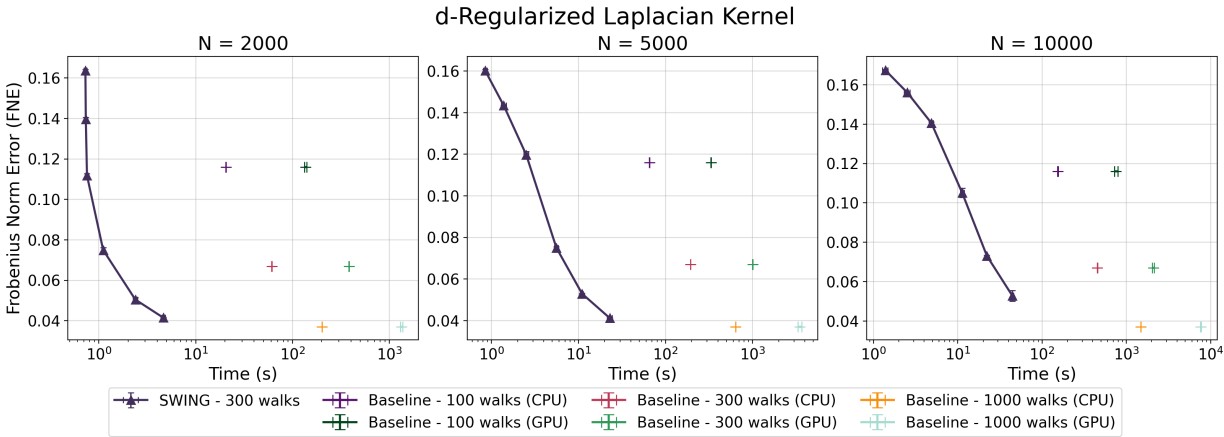

*Figure 10.* Trade-off between approximation error (FNE) and computation time for the $d-$regularized laplacian kernel on synthetic point clouds of varying size.

Figures 7 and 8 report the kernel approximation error as a function of the number of random features for the diffusion kernel and the $d$-regularized Laplacian kernel, respectively, across increasing point cloud sizes. As in the $p$-step random walk case, the approximation error decreases monotonically as the number of random features increases. For a fixed number of random walks, the proposed method consistently attains lower approximation error than the GRF baseline for a suitable range of random feature dimensions.

Figures 9 and 10 show the corresponding trade-off between approximation error and computation time. In both cases, the proposed method traces an empirical Pareto frontier, allowing accuracy–time trade-offs to be adjusted smoothly by varying the number of random features. The same qualitative behavior observed for the $p$-step random walk kernel holds here as well, with substantial inference-time speedups compared to the baseline methods.

## D. Additional Experimental Details

In this section, we will provide additional experimental details for experiments of Section 3.2.

### D.1. Vertex Normal Prediction

In this sub-section, we present implementation details for vertex normal prediction experiments. All the experiments are run on free Google Colab with 12Gb of RAM on CPU.

*Table 2.* Cosine Similarity for Meshes. SWING performs at par as GRF while being more computationally efficient. We also compare against Al Mohy's algorithm to compute the action of the matrix exponential on a vector. BF refers to the brute force version where the kernel is explicitly materialized followed by matrix-vector multiplication.

| MESH SIZE | 64 | 99 | 146 | 148 | 155 | 182 | 222 | 246 | 290 | 313 |
|---|---|---|---|---|---|---|---|---|---|---|
| BF | 0.8461 | 0.8361 | 0.8210 | 0.6308 | 0.6604 | 0.9702 | 0.8913 | 0.4280 | 0.5138 | 0.3221 |
| Al Mohy | 0.8457 | 0.8225 | 0.8210 | 0.6196 | 0.6493 | 0.9702 | 0.8904 | 0.4277 | 0.5075 | 0.3087 |
| Lanczos | 0.7855 | 0.8033 | 0.6797 | 0.4504 | 0.4994 | 0.9403 | 0.4866 | 0.2260 | 0.3438 | 0.2207 |
| GRF | 0.8255 | 0.8266 | 0.8321 | 0.5643 | 0.5869 | 0.9805 | 0.8343 | 0.4403 | 0.4852 | 0.3105 |
| SWING (ours) | 0.8301 | 0.8275 | 0.8197 | 0.5667 | 0.6226 | 0.9574 | 0.8226 | 0.4264 | 0.5145 | 0.3089 |

| MESH SIZE | 362 | 482 | 502 | 518 | 614 | 639 | 777 | 942 | 992 | 1012 |
|---|---|---|---|---|---|---|---|---|---|---|
| BF | 0.7725 | 0.9883 | 0.8710 | 0.1363 | 0.6163 | 0.9102 | 0.7135 | 0.7890 | 0.8501 | 0.7158 |
| Al Mohy | 0.7588 | 0.9844 | 0.8693 | 0.1170 | 0.6129 | 0.9077 | 0.7068 | 0.7633 | 0.8401 | 0.7150 |
| Lanczos | 0.1466 | 0.1144 | 0.2536 | 0.1613 | 0.0384 | 0.1531 | 0.1271 | 0.6593 | 0.7014 | 0.7114 |
| GRF | 0.7567 | 0.9497 | 0.7154 | 0.1172 | 0.6266 | 0.9024 | 0.6647 | 0.7681 | 0.8354 | 0.6932 |
| SWING (ours) | 0.7386 | 0.9346 | 0.6872 | 0.0916 | 0.6136 | 0.8847 | 0.6203 | 0.7300 | 0.8336 | 0.6998 |

| MESH SIZE | 1094 | 1192 | 1849 | 2599 | 2626 | 2996 | 3072 | 3559 | 3715 | 4025 |
|---|---|---|---|---|---|---|---|---|---|---|
| BF | 0.6765 | 0.6737 | 0.9214 | 0.7601 | 0.8567 | 0.8864 | 0.6240 | 0.8090 | 0.8061 | 0.7221 |
| Al Mohy | 0.6581 | 0.6683 | 0.9139 | 0.7500 | 0.8503 | 0.8743 | 0.6228 | 0.7925 | 0.7998 | 0.7214 |
| Lanczos | 0.0768 | 0.4463 | 0.3487 | 0.4019 | 0.3847 | 0.6037 | 0.2541 | 0.1262 | 0.2156 | 0.1812 |
| GRF | 0.6280 | 0.6313 | 0.8701 | 0.7296 | 0.8427 | 0.8021 | 0.5726 | 0.7529 | 0.7819 | 0.7131 |
| SWING (ours) | 0.6255 | 0.6224 | 0.8401 | 0.7067 | 0.8317 | 0.8086 | 0.5332 | 0.7483 | 0.7664 | 0.7152 |

| MESH SIZE | 5155 | 5985 | 6577 | 6911 | 7386 | 7953 | 8011 | 8261 | 8449 | 8800 |
|---|---|---|---|---|---|---|---|---|---|---|
| BF | 0.7927 | 0.8133 | 0.8960 | 0.9306 | 0.8161 | 0.7991 | 0.7553 | 0.7153 | 0.8090 | 0.9361 |
| Al Mohy | 0.7927 | 0.8028 | 0.8801 | 0.9196 | 0.8026 | 0.7816 | 0.7387 | 0.7013 | 0.7899 | 0.9257 |
| Lanczos | 0.5186 | 0.7149 | 0.6113 | 0.7157 | 0.1435 | 0.4093 | 0.5985 | 0.0718 | 0.6772 | 0.5130 |
| GRF | 0.7749 | 0.7947 | 0.7730 | 0.9022 | 0.7965 | 0.7743 | 0.7254 | 0.6894 | 0.8019 | 0.9201 |
| SWING (ours) | 0.7532 | 0.7731 | 0.8484 | 0.8399 | 0.7890 | 0.7650 | 0.6975 | 0.6798 | 0.7594 | 0.9137 |

For this task, we choose 40 meshes for 3D-printed objects of varying sizes from the Thingi10K dataset. Following Choromanski et al. (2023), we choose the following meshes corresponding to the ids given by :

```
[60246, 85580, 40179, 964933, 1624039, 91657, 79183, 82407, 40172, 65414, 90431,
74449, 73464, 230349, 40171, 61193, 77938, 375276, 39463, 110793, 368622, 37326,
42435, 1514901, 65282, 116878, 550964, 409624, 101902, 73410, 87602, 255172,
98480, 57140, 285606, 96123, 203289, 87601, 409629, 37384, 57084]
```

For the baseline, we use a kernel of the form $\mathbf{K} = \exp(\lambda \mathbf{W})$, where $\mathbf{W} = e^{-\frac{||x_i - x_j||^2}{\sigma^2}}$. We do a small search for the $\sigma$ and $\lambda$ for the baseline kernel. We also compare against Al Mohy and Lanczos (Musco et al., 2018) method as they are widely popular algorithms to compute the action of matrix exponentials on a vector. We use the SciPy implementation of Al Mohy. For both GRF and SWING, the number of walks are chosen from the subset $\{16, 32, 64, 128, 256, 300\}$ and the halting probability of the walk is $0.1$. Additionally we do further hyperparameter tuning on the number of random features which is chosen from $\{8, 16, 32, 64, 128, 256, 512\}$.

Finally we show runtime comparison of SWING against the explicit kernel construction (BF) and GRF. SWING scales better than both BF and GRF while performing similarly (Fig. 11). For this runtime experiment, we fix the number of random features to be 128, number of walks to be 200, halting probability to be 0.15. The $\sigma$ and $\lambda$ are same as in the previous experiment.

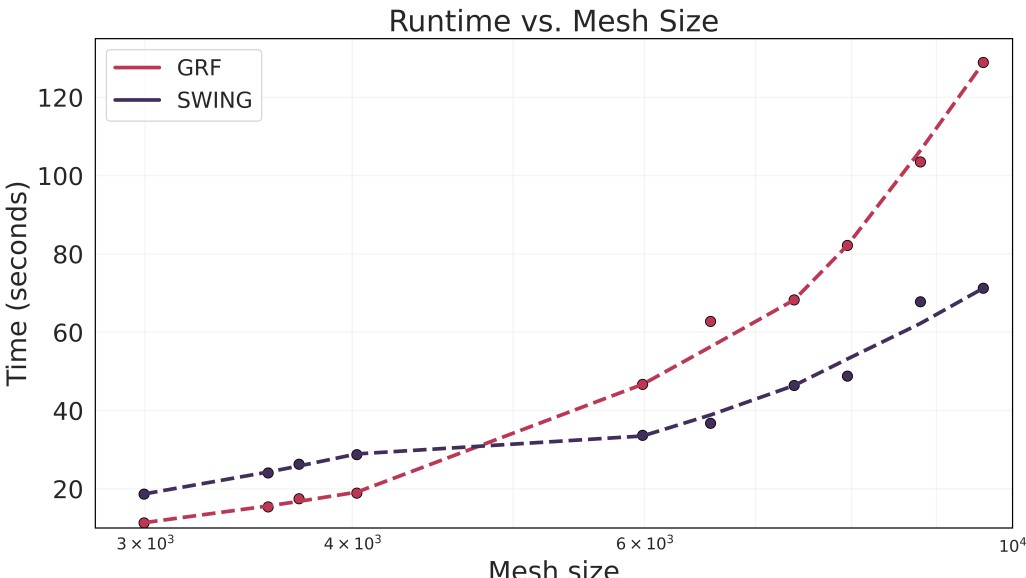

*Figure 11.* Runtime on CPU to create the kernel matrix as a function of mesh size. SWING performs faster than both the explicit kernel construction method as well as GRF as the mesh size grows.

