# OpenReview forum: "SWING: Unlocking Implicit Graph Representations for Graph Random Features"
_ICML.cc/2026/Conference — ICML 2026 spotlight_

### Official Review · Reviewer_ZPDa · 2026-03-07

**Soundness:** 3
**Presentation:** 2
**Significance:** 3
**Originality:** 4
**Overall Recommendation:** 4
**Confidence:** 3

**Summary:**

This paper presents a speed-up technique of conducting random walks over implicit graphs (the authors called them i-graphs). Compared with the random walk over explicit graphs, the random walk over implicit graph may be required when the graphs are hard to construct or too costly to construct. The authors give a technique to effectively approximate the GRF algorithm, and run random walks over R^d, the feature space of nodes.

The technical contributions seem solid, yet there may be limitations on the experimental results.

**Compliance With Llm Reviewing Policy:**

Affirmed.

**Final Justification:**

The topic is interesting and techniques are non-trivial and hence I would like to maintain my positive score.

**Key Questions For Authors:**

Q1. If I understand correctly, the strength of the proposed algorithm is that running one random walk is in O(N) time, faster than the original GRF algorithm. However, running a random walk in O(N) time does not seem to be very efficient if compared with random walks over explicit graphs. The question is, why not construct a graph, and compare with the random walk efficiency over explicit graphs?

Q2. What do you mean by O_N(1). It appears several times.

Q3. In experiments (Fig. 2), the error of SWING goes down with the increase of the number of random features. Typically, SWING excels the baseline (300 walks) when the number of features exceed N. Is there any property here? Also, based on this property, I worry that the SWING method may only be more useful when N is small. However, in cases where N is small, the benefit of speed-ups diminished.

**Limitations:**

I believe there are some novelty on the technical contributions, yet the experimental evaluation has limitations (see questions above).

**Strengths And Weaknesses:**

S1. This is my first time reading the algorithm of random walks over i-graphs. They are novel and interesting.

S2. The introduction and background is pretty well presented.

S3. Although I have not thoroughly checked the details, the techniques seem non-trivial.

W1. The presentation of the technical sections need improvement.

W2. The experimental evaluation is not sufficient.

---

> ### Author Rebuttal · Authors · 2026-03-31
>
> We thank the Reviewer for the constructive feedback.
>
> **W1**
>
> Following the Reviewer's comment, we will further improve the technical section by providing an algorithm box with pseudocode describing SWING. We will also provide a more detailed discussion of various classes of random feature mechanisms used to approximate kernels, with emphasis on particularly efficient techniques leveraging positive random features.
>
> **W2**
>
> Following the Reviewer's comment, we have added additional experimental results confirming the strong performance of SWING. In particular, we include the well-known Lanczos method [1] in the experiments from Sec. 3.2, with 40 meshes. The comparison of SWING with Lanczos is given below.
>
> | Mesh size | 5155 | 5985 | 6577 | 6911 | 7386 | 7953 | 8011 | 8261 | 8449 | 8800 |
> |---|---:|---:|---:|---:|---:|---:|---:|---:|---:|---:|
> | Lanczos | 0.5186 | 0.7149 | 0.6113 | 0.7157 | 0.1435 | 0.4093 | 0.5985 | 0.0718 | 0.6772 | 0.513 |
> | SWING (ours) | 0.7532 | 0.7731 | 0.8484 | 0.8399 | 0.789 | 0.765 | 0.6975 | 0.6798 | 0.7594 | 0.9137 |
>
> SWING outperforms the Lanczos on all meshes.
>
> For ViT methods, we added the comparison with the ViT-GRF variant, in addition to the comparison with regular ViT (ViT-GRF is considered an attention-efficient method). The updated results are presented below.
>
> |  | ImageNet | Places365 |
> |---|---:|---:|
> | ViT | 80.10 | 55.78 |
> | ViT-GRFs | 80.09 | 55.39 |
> | ViT-SWING (ours) | 80.34 | 55.97 |
>
> ViT-SWING provides the best performance on both datasets. We conjecture that SWING's continuous relaxation, as a by-product, provides useful regularization from which ViT models benefit.
>
> In the vision domain, gains of this size are routinely reported for RPE-like additions; see, e.g., Table 1 in [3], which reports gains of **0.14\%** and **0.18\%** for a transformer using relative positional encoding (RPE) trained on ImageNet and Places365, respectively.
>
> **Q1**
>
> Thank you for the comment. Let us clarify our setting. SWING replaces walks on the nodes of the graph with walks in the continuous space in which those graphs are embedded. The computational improvement comes from the fact that walks in continuous space can be computed using the attention mechanism, much more efficiently than walks on the nodes of the graph. SWING uses short walks of constant average length, rather than walks of length linear in $N$ (note that, as stated in line 311, $p_{\mathrm{halt}} = 0.3$). Thus, SWING inherits the short-walk approach from regular GRFs. The comparison with the baseline in Figs. 2 and 3 is a comparison with the regular GRF method, where graphs are explicitly materialized and walks on the nodes of the graph are explicitly computed. As shown in Fig. 3, the explicit approach is much less computationally efficient. More specifically, **SWING achieves inference speedups exceeding one order of magnitude**.
>
>
> **Q2**
>
> We use this notation for time complexity that does not depend on $N$ (the number of nodes of the i-graph). We will clarify it in the paper.
>
> **Q3**
>
> Thank you for your comment. With the best bounded positive random feature mechanisms for the kernels considered in this paper, an analysis similar to that conducted in the proof of Claim 1 of [2] can be undertaken. In particular, for any given $\epsilon > 0$, the probability that the approximation error exceeds $\epsilon$ for at least one pair of points for which kernel values are calculated is upper-bounded by an expression proportional to $\exp\left(-\frac{n_{\mathrm{rf}}}{d}\right) \cdot \left(\frac{\mathrm{diam}(\mathcal{M})}{\epsilon}\right)^{2}$, where $n_{\mathrm{rf}}$ is the number of RFs, $d$ is the dimension of the space where the points are embedded (in the presented applications, $d = 3$), and $\mathrm{diam}(\mathcal{M})$ is the diameter of the region where all points live. This bound does not depend on the number of points. Consequently, the number $n_{\mathrm{rf}}$ of RFs needed for a particular error level $\epsilon$ with high probability (simultaneously for all pairs of points) grows as $\frac{d}{\epsilon^{2}}\log(\frac{\mathrm{diam}(\mathcal{M})}{\epsilon})$ and **is independent of the number of points $N$**. Following the Reviewer's comment, we will provide the corresponding detailed analysis in the final version of the paper.
>
> It should also be emphasized that even if a larger number of random features is used in SWING, the corresponding calculations can be conducted very efficiently on accelerators, as compared to regular graph-node walk calculations in standard GRFs. As mentioned above, this is clearly illustrated in Fig. 3, where SWING can achieve high accuracy with 10x+ inference speedups, as compared to GRFs.
>
> [1] Stability of the Lanczos Method for Matrix Function Approximation; Musco et al.; 2018
>
> [2] Random Features for Large-Scale Kernel Machines; Rahimi & Recht; 2007
>
> [3] Learning the RoPEs: Better 2D and 3D Position Encodings with STRING; Schenck et al.; 2025

---

> > ### Author Rebuttal · Reviewer_ZPDa · 2026-04-03
> >
> > Thanks for the rebuttal. I have no more questions and will keep my positive rating.

---

### Official Review · Reviewer_Gphs · 2026-03-11

**Soundness:** 3
**Presentation:** 3
**Significance:** 3
**Originality:** 3
**Overall Recommendation:** 5
**Confidence:** 3

**Summary:**

This paper proposes an efficient method for approximating Graph Random Features (GRF) specifically for implicit graphs, where edge weights are defined by the differences between node features. The proposed approach utilizes continuous relaxation of the GRF via the Gumbel-Softmax trick and random features for the approximation. This architecture ensures that both the graph construction and the subsequent computation of random features scale linearly with the number of nodes ($O(N)$). The authors validate the effectiveness of their method through experiments on synthetic data, a 3D object normal prediction task, and performance enhancements in ViT models.

**Compliance With Llm Reviewing Policy:**

Affirmed.

**Ethical Review Concerns:**

N.A.

**Final Justification:**

My questions in my review has been resolved. So, I keep my score, leaning toward acceptance.

**Key Questions For Authors:**

1. Regarding the ViT experiments, could you provide an ablation study or further analysis to isolate the performance gains attributable to the SWING mechanism from those provided by the soft-mask-based regularization?

**Limitations:**

yes

**Strengths And Weaknesses:**

Strengths
Soundness
- The proposed method is built upon a solid mathematical foundation, and the theoretical derivations provided are well-justified.
- Empirical results clearly demonstrate that the approximation accuracy improves consistently as the number of random features increases, supporting the technical claims of the paper.
- The experiments show that the proposed approach achieves better approximation accuracy than Gaussian Random Features (GRF) as the number of features increases, effectively showcasing its empirical superiority and practical utility.

Presentation
- The paper is well-written, clearly structured, and easy to follow. The mathematical notation and descriptions are precise, making the theoretical components accessible to the reader.
- The work is properly positioned within the context of existing literature; specifically, the relationship with Gaussian Random Features (GRF) is well-articulated, and the comparative advantages of the proposed method are clearly justified.

Significance
- The practical impact of this work is substantial. As demonstrated in Figure 3, the proposed method achieves a significant speedup—by dozens of times—to reach the same level of approximation error compared to existing baselines. This level of efficiency gain is highly impactful for large-scale graph analysis, where computational cost is often a primary bottleneck, making the proposed approach very relevant for real-world applications.

Originality
- Although the individual components—namely random features and the Gumbel trick—are relatively standard techniques in the fields of kernel acceleration and discrete sampling, to the best of my knowledge, their integration within the context of computing random features on graphs is novel.


Weaknesses
Soundness
- In the ViT experiments, the proposed method incorporates both soft-mask-based regularization and SWING approximation. Consequently, the individual contribution of the SWING mechanism compared with GRF is not clearly isolated from the effects of the regularization.

---

> ### Author Rebuttal · Authors · 2026-03-31
>
> We appreciate the valuable constructive feedback and respond to all comments below.
>
> **ViT experiments**
>
> Thank you very much for the comments. Let us clarify the setting used there. Our base model is a regular ViT with standard softmax attention. The ViT-SWING model is a modification of the regular ViT in which the standard attention matrix is modulated by a mask matrix, whose entries are given by the SWING relaxation of the approximate GRF-based graph kernel matrix. Thus, the only difference between the ViT and ViT-SWING settings is the SWING-based mask matrix. The corresponding accuracy improvements are provided in Table 1. These clearly quantify the sole impact of SWING. **To further disentangle the gains provided by SWING** from those that the ViT-GRF model would provide (SWING is the continuous relaxation of GRFs), **for the rebuttal we also trained the ViT-GRF model**. The comparison of all three models, ViT, ViT-SWING, and ViT-GRF, is provided below:
>
> |  | ImageNet | Places365 |
> |---|---:|---:|
> | ViT | 80.10 | 55.78 |
> | ViT-GRF | 80.09 | 55.39 |
> | ViT-SWING (ours) | 80.34 | 55.97 |
>
> **ViT-SWING provides the best performance** on both ImageNet and Places365. We conjecture that SWING's continuous relaxation, as a by-product, provides useful regularization, compared to ViT-GRF, from which ViT models benefit.
>
> In the vision domain, gains of this size are routinely reported for RPE-like additions; see, e.g., Table 1 in [1], which reports gains of 0.**14\%** and **0.18\%** for a transformer using relative positional encoding (RPE) trained on ImageNet and Places365, respectively.
>
> [1] Learning the RoPEs: Better 2D and 3D Position Encodings with STRING; Schenck et al.; 2025

---

> > ### Author Rebuttal · Reviewer_Gphs · 2026-04-03
> >
> > I appreciate the authors for answering my questions. I am satisfied with the comparison with ViT-GRF and ViT-SWING.

---

### Official Review · Reviewer_ZDxy · 2026-03-13

**Soundness:** 2
**Presentation:** 2
**Significance:** 3
**Originality:** 4
**Overall Recommendation:** 5
**Confidence:** 2

**Summary:**

The paper proposes SWING, a novel algorithm designed to perform computations involving graph random features (GRFs) on implicit graphs without the need for computationally expensive graph materialization. SWING replaces discrete walks for standard GRFs with continuous spatial random walks in the embedding space, and thus reduce complexity for graphs defined by pairwise functions of node features. This is achieved via a Gumbel-softmax sampling mechanism combined with linearized kernels obtained through random features and importance sampling. The authors provide a theoretical derivation for these "space walks" and demonstrate through experiments on synthetic point clouds, 3D meshes, and Vision Transformers that SWING achieves significant computational speedups, while maintaining approximation accuracy comparable to exact methods.

**Compliance With Llm Reviewing Policy:**

Affirmed.

**Final Justification:**

I believe the paper can be accepted after the appropriate update.

**Key Questions For Authors:**

The continuous relaxation relies on the Gumbel-softmax temperature. How sensitive is the approximation accuracy to this hyperparameter?

Would the bias introduced by the Gumbel relaxation accumulate as the walk length T increases, particularly in comparison with the unbiased discrete random walks employed in standard Generalized Random Features (GRFs)?

**Limitations:**

The authors have briefly touched upon the impact statement, noting they do not feel specific societal consequences must be highlighted. The paper would benefit from a more explicit discussion on the limitations of the "space walk" approximation for scenarios where the continuous relaxation might fail to capture complex topological structures that a discrete walk would find.

**Strengths And Weaknesses:**

The problem of computing graph-level features on implicit graphs without materializing them is relevant and practically important, especially in domains like point cloud processing and attention-based architectures. The authors clearly derive the transition from discrete graph walks to continuous space walks using the Gumbel-Trick and random feature linearization.

The method is fundamentally designed for implicit graphs. This severely limits its application in general Graph Representation Learning, as it cannot be directly applied to explicit graphs with arbitrary topologies. The work employs a chain of approximations consisting of random Fourier features and Gumbel-softmax relaxation, which introduces multiple sources of error. Although each approximation is individually justified, a unified end-to-end error bound or convergence rate analysis is not provided. There is a risk that approximation errors may compound as the length of the random walk increases, potentially making the method unstable or difficult to tune compared to exact discrete methods, especially in high-dimensional settings.

---

> ### Author Rebuttal · Authors · 2026-03-31
>
> We thank the Reviewer for their time and feedback, which have significantly helped improve our work. Our point-by-point responses are detailed below.
>
> **Implicit graph focus**
>
> Implicit graphs constitute a prominent class of networks considered in ML, in particular for the point cloud (PC) data modality, where graph-based modeling is often applied. $\epsilon$-neighborhood graphs are among the most popular ways of graph modeling for PCs, and these are examples of implicit networks. The importance of implicit graphs in ML comes from the fact that, very often, the strength of the connection between two given nodes can be written as a (potentially complicated) function of the feature vectors defining the properties of those nodes. Such graphs are also implicit. Implicit graphs are also very popular in ML because they can be efficiently represented for calculations involving accelerators via dense tensors of feature vectors (representing nodes), in space linear in the number of nodes $N$, regardless of whether they are sparse or dense. This is not the case for graphs defined explicitly. Such graphs, if dense, need to be represented as adjacency matrices, occupying quadratic space in $N$. They can thus be processed efficiently only for relatively small values of $N$. Explicitly defined sparse graphs can be represented in linear space via adjacency lists, but this data structure is not optimized for computations on accelerators. There exist libraries for processing sparse adjacency matrices on accelerators, but their computational efficiency is much lower than that of computations involving dense tensors. For that reason, $\epsilon$-neighborhood graphs are often chosen over meshes to model PC data.
>
> Having said that, following the Reviewer's comment, in the camera-ready version we will add a Limitations section clarifying that SWING is designed for implicitly, rather than explicitly, defined graphs. In that section, we will also explain that the continuous relaxation of the discrete mechanism of regular walks on graph nodes is a computational-efficiency versus accuracy trade-off.
>
> **Error analysis**
>
> Thank you very much for this important question. An empirical answer is provided by Figs. 2, 7, and 8, where the Frobenius norm error of SWING converges to zero and is competitive with that of regular GRFs. For completeness, let us also provide a sketch of the theoretical analysis. Following the Reviewer's comment, we will add the full analysis to the camera-ready version.
>
> There are a couple of sources of approximation error, as the Reviewer emphasized:
>
> 1) error of the kernel value approximation,
> 2) error of the approximation of sampling from the categorical distribution,
> 3) accumulating error over steps of the walk.
>
> Let us analyze these sources one by one:
>
> 1) With the best bounded positive random feature mechanisms for the kernels considered in this paper, an analysis similar to that conducted in the proof of Claim 1 of [1] can be undertaken. In particular, for any given $\epsilon > 0$, the probability that the approximation error exceeds $\epsilon$ for at least one pair of points for which kernel values are calculated is upper-bounded by an expression proportional to $\exp\left(-\frac{n_{\mathrm{rf}}}{d}\right) \cdot \left(\frac{\mathrm{diam}(\mathcal{M})}{\epsilon}\right)^2$, where $n_{\mathrm{rf}}$ is the number of RFs, $d$ is the dimension of the space where the points are embedded (in the presented applications, $d = 3$), and $\mathrm{diam}(\mathcal{M})$ is the diameter of the region where all points lie. This bound does not depend on the number of points. Consequently, the number $n_{\mathrm{rf}}$ of RFs needed for a particular error level $\epsilon$ with high probability (simultaneously for all pairs of points) grows as:
> $\frac{d}{\epsilon^{2}}\log(\frac{\mathrm{diam}(\mathcal{M})}{\epsilon})$.
>
> 2) For bounded feature vectors, standard tools can also be applied to uniformly upper-bound the approximation error introduced by the Gumbel-softmax trick. Note that here we also apply softmax kernel approximation with positive random features, and the analysis from (1) can again be applied.
>
> 3) It is indeed the case that, in principle, the error accumulates over the length of the walk $T$. However, in practice $T$ can be taken to be a small constant, and thus this error accumulation is not detrimental. Indeed, previous papers on GRFs (Sec. 5.2 in [2] and Secs. 3.2, 3.3, and 3.4 in [3]) showed that $T$ between 2 and 10 is sufficient to provide a good approximations of graph kernel values.
>
> **Sensitivity to temperature hyperparameter**
>
> The standard hyperparameter interval $[0.1, 1]$, usually considered for Gumbel-softmax, also works for SWING. We will clarify this in the camera-ready version.
>
> [1] Random Features for Large-Scale Kernel Machines; Rahimi & Recht; 2007
>
> [2] Taming graph kernels with random features; Choromanski, Krzysztof M.; 2023.
>
> [3] General Graph Random Features; Reid, Isaac, et al.; 2025

---

> > ### Author Rebuttal · Reviewer_ZDxy · 2026-04-04
> >
> > Thank you for the detailed response. I am updating my score accordingly and encourage the authors to include the analysis in the camera-ready version of the paper.

---

### Official Review · Reviewer_qKD8 · 2026-03-17

**Soundness:** 3
**Presentation:** 3
**Significance:** 4
**Originality:** 3
**Overall Recommendation:** 5
**Confidence:** 4

**Summary:**

This paper proposes SWING, a method for performing Graph Random Feature (GRF)-based kernel computations on implicit graphs, where edge weights are defined by a function of node features rather than by an explicitly materialized adjacency matrix. The key idea is to replace discrete random walks on graph nodes with continuous-space walks in the embedding space, using a Gumbel-softmax relaxation, and then use random features and Fourier analysis to derive a linear-time-in-\(N\) approximation of kernel-vector multiplication without explicitly constructing the graph.

The paper is motivated by the observation that existing graph kernel methods can be computationally very expensive, and that standard GRF approaches still require explicit graph construction and discrete graph walks. SWING is intended to address this gap by operating directly on implicit graph representations. The method is evaluated on synthetic 3D point clouds, mesh normal prediction, and ViT-based image classification.

**Compliance With Llm Reviewing Policy:**

Affirmed.

**Key Questions For Authors:**

- In the ViT experiment, the gains are fairly small. How stable are these improvements across seeds?
- How does the approximation error scale with respect to the dimensionality $d$ of the embedding space?

**Limitations:**

yes

**Strengths And Weaknesses:**

Strengths:

- The paper addresses a meaningful and practically relevant limitation of current GRF methods: the need to explicitly construct graphs before performing random walks. This is especially important for geometric settings such as point clouds or \(\epsilon\)-neighborhood graphs.

- By removing combinatorial operations and replacing them with attention-like continuous relaxations, SWING is explicitly designed to maximize the parallel processing power of modern accelerators like GPUs.

- The empirical validation is thorough, spanning synthetic validations across multiple kernel types and meaningful downstream tasks like point cloud modeling and image classification.

Weaknesses:
- The method relies heavily on specific hyperparameter choices, such as the Gumbel temperature, the number of random features, and the scaling parameter. The paper lacks a detailed ablation study showing how sensitive the model is.

- For the 3D mesh and ViT experiments, SWING is primarily compared against standard GRFs and brute-force explicit materialization. It would be helpful to this paper if more methods are included.

---

> ### Author Rebuttal · Authors · 2026-03-31
>
> We would like to sincerely thank the Reviewer for the insightful feedback. We address all questions below.
>
> **Hyperparameters**
>
>  Thank you for this comment. We conducted different grid searches to select the final hyperparameters, as partially detailed in Appendix D, and we will add a more in-depth description of the hyperparameter search procedure in the final version of the paper.
>
> For the Gumbel temperature, values from the standard optimization interval $[0.1, 1]$ worked well. For the scaling parameter, we applied a log scale in $[0.1, 10] \cdot N^{1/3}$. As the optimization landscape is convex, faster convergence to optimal hyperparameters can be achieved with Bayesian search methods (e.g., TPE).
>
> Increasing the number of random features $n_{\mathrm{rf}}$ improves accuracy, since it plays the role of the number of samples in the Monte Carlo approximation. Note that in our experiments (Figs. 3, 9, and 10), there always exists a threshold value of $n_{\mathrm{rf}}$ beyond which we obtain a better kernel approximation together with a substantial speed-up.
>
> We would also like to note that, for points taken from a bounded region and the most efficient random-feature mechanisms relying on bounded features (which exist for all kernels considered in SWING), the $n_{\mathrm{rf}}$ required to obtain a given level of kernel-approximation accuracy **simultaneously for all pairs of points** does not depend on the number of points, but only on the desired error level and the radius of the ball containing all points. This follows from an analysis analogous to that conducted in [1] (see Claim 1 and its corollary), leveraging standard concentration inequalities (e.g., Chernoff bounds) and an $\epsilon$-net argument.
>
> To summarize, the method itself is not particularly sensitive to the choice of hyperparameters. However, we agree with the Reviewer that the paper would benefit from a more detailed discussion of this aspect, and we will add it to the camera-ready version.
>
> **Additional methods**
>
> Following the Reviewer's comment, we provide additional experimental results confirming SWING's strong performance. In particular, we include the well-known Lanczos method [2] in the experiments from Sec. 3.2, using 40 meshes. The comparison between SWING and Lanczos is given below.
>
> | Mesh size | 5155 | 5985 | 6577 | 6911 | 7386 | 7953 | 8011 | 8261 | 8449 | 8800 |
> |---|---:|---:|---:|---:|---:|---:|---:|---:|---:|---:|
> | Lanczos | 0.5186 | 0.7149 | 0.6113 | 0.7157 | 0.1435 | 0.4093 | 0.5985 | 0.0718 | 0.6772 | 0.513 |
> | SWING (ours) | 0.7532 | 0.7731 | 0.8484 | 0.8399 | 0.789 | 0.765 | 0.6975 | 0.6798 | 0.7594 | 0.9137 |
>
> SWING outperforms the Lanczos method on all **10 mesh sizes**.
>
> For ViT-based methods, we added a comparison with the ViT-GRF variant, in addition to the comparison with the standard ViT. The updated results are presented below.
>
> | Method | ImageNet | Places365 |
> |---|---:|---:|
> | ViT | 80.10 | 55.78 |
> | ViT-GRFs | 80.09 | 55.39 |
> | ViT-SWING (ours) | 80.34 | 55.97 |
>
> ViT-SWING achieves the best performance on both ImageNet and Places365. We conjecture that SWING's continuous relaxation, as a by-product, provides useful regularization compared to ViT-GRFs, from which ViT models benefit. In the vision domain, gains of this magnitude are routinely reported for RPE-like additions; see, e.g., Table 1 in [3], which reports gains of **$0.14\%$** and **$0.18\%$** for a transformer with relative positional encoding (RPE) trained on ImageNet and Places365, respectively.
>
> **ViT improvements across seeds**
>
>  Thank you for this question. The gains were consistent across **five** different runs. We will clarify this in the camera-ready version. As noted above, in the vision domain, gains of this magnitude are routinely reported for RPE-like additions.
>
> **Approximation error as a function of $d$**
>
> Thank you for this excellent question. With the best bounded positive random feature mechanisms for the kernels considered in this paper, similar analysis to the one conducted in the proof of Claim 1 of [1] can be carried out. In particular, for any given $\epsilon > 0$, the probability that the approximation error exceeds $\epsilon$ for at least one pair of points for which kernel values are computed is upper-bounded by an expression proportional to
> $\exp(-\frac{n_{\mathrm{rf}}}{d}) \cdot (\frac{\mathrm{diam}(\mathcal{M})}{\epsilon})^{2}$,
> where $n_{\mathrm{rf}}$ is the number of random features, $d$ is the dimension of the space in which the points are embedded (in the applications presented here, $d = 3$), and $\mathrm{diam}(\mathcal{M})$ is the diameter of the region containing all points. We will clarify this in the camera-ready version.
>
>
> [1] Random Features for Large-Scale Kernel Machines; Rahimi & Recht; 2007
>
> [2] Stability of the Lanczos Method for Matrix Function Approximation; Musco et al.; 2018
>
> [3] Learning the RoPEs: Better 2D and 3D Position Encodings with STRING; Schenck et al.; 2025

---

> > ### Author Rebuttal · Reviewer_qKD8 · 2026-04-03
> >
> > Thank you for the rebuttal. The authors have clarified my concerns. I will maintain my positive score.

---

### Decision · Program_Chairs · 2026-04-30

**Decision:**

Accept (spotlight)

**Comment:**

The paper introduces SWING, a method that replaces discrete graph operations with continuous, attention-like relaxations, enabling better parallelization on GPUs, in order to tackle a key limitation of Graph Random Features (GRF) methods, i.e., the need to explicitly construct graphs before performing random walks, which is inefficient in geometric settings like point clouds.
The proposed approach is theoretically sound, with clear derivations using techniques like the Gumbel trick and random feature linearization. Empirical results show that SWING consistently improves approximation accuracy as the number of features increases and outperforms traditional GRF methods.
The paper is well-written and clearly situates itself within existing literature. Its main contribution lies in efficiently computing graph-level features on implicit graphs without explicitly building them, leading to major speed improvements, making the proposed approach actionable for real-world applications involving large-scale graphs.
Overall, the work is practically significant and novel in how it combines established techniques (random features and the Gumbel trick) within this new context.
All the issues raised by the reviews were resolved during the rebuttal.